## RESEARCH ARTICLE

# Spatiotemporal dynamics of ecto-5′-nucleotidase (CD73) in mouse retina under physiological conditions

Ryutaro Ishii[1,2,*], Keisuke Sakurai[3], Nao Hosomi[2,4], Bernd K. Fleischmann[5], Seiya Mizuno[1,6], Kenichi Kimura[2,*] and Hiromi Yanagisawa[2]

## ABSTRACT

Adenosine is essential for energy metabolism and acts as a neuromodulator in the central nervous system. The retina is a highly energy-demanding neural tissue, and dysregulation of adenosine metabolism contributes to retinal diseases. Although adenosine pathways form extensive compensatory networks, the dynamics of ecto-5′-nucleotidase (CD73), a key enzyme for extracellular adenosine generation, remain elusive. Here, we map its spatiotemporal profile from development to adulthood using two transgenic mouse lines. We found that CD73 became expressed in the rod-photoreceptor lineage by postnatal day (P) 3 and appeared in the inner nuclear layer from P7 onward. Moreover, CD73 was transiently expressed in the astrocyte lineage between embryonic day 16.5 and P3, and the descendants accumulated in the retinal periphery. Functionally, CD73 deletion delayed rod-response recovery and shortened the implicit time in scotopic electroretinography under dim light. These findings suggest that physiological cues such as light and hypoxia may influence CD73 expression and retinal function. Our work lays the groundwork for investigating how genetic and environmental risk factors may alter adenosine metabolism in retinal diseases.

KEY WORDS: Adenosine, Ecto-5′-nucleotidase (CD73), Light/dark adaptation, Lineage tracing, Mouse retina, Retinal development

## INTRODUCTION

Adenosine is essential to cellular energy metabolism as a component of adenosine triphosphate (ATP) and a potent extracellular neuromodulator that shapes neuronal excitability, transmitter release and the sleep–wake cycle (Garcia-Gil et al., 2021; Lazarus et al., 2019; Wei et al., 2011). Extracellular adenosine is produced mainly by ecto-5′-nucleotidase (CD73), which hydrolyzes adenosine monophosphate (AMP) to adenosine, and is cleared

[1]Institute of Medicine, University of Tsukuba, Tsukuba 305-8575, Japan. [2]Life Science Center for Survival Dynamics, Tsukuba Advanced Research Alliance (TARA), University of Tsukuba, Tsukuba 305-8577, Japan. [3]Institute of Life and Environmental Sciences, University of Tsukuba, Tsukuba 305-8572, Japan. [4]Doctoral Program in Human Biology, Comprehensive Human Sciences Research Group, Graduate School of Comprehensive Human Sciences, University of Tsukuba, Tsukuba 305-0006, Japan. [5]Institute of Physiology I, Medical Faculty, University of Bonn, Bonn 53127, Germany. [6]Laboratory Animal Resource Center, Transborder Medical Research Center, Institute of Medicine, University of Tsukuba, Tsukuba 305-8575, Japan.

*Authors for correspondence (rtrishii-tuk@umin.ac.jp; kkimura@tara.tsukuba.ac.jp)

R.I., 0000-0002-7919-6354; B.K.F., 0000-0002-9202-8363; S.M., 0000-0002-6740-5817; K.K., 0000-0002-0363-8865; H.Y., 0000-0002-7576-9186

when intracellular adenosine kinase (ADK) re-phosphorylates adenosine imported through equilibrative nucleoside transporters (ENTs). Because genetic deletion of adenosine signaling components often triggers compensatory pathways (Garcia-Gil et al., 2021; Wei et al., 2011), dissecting adenosine pathways demands tools with high spatiotemporal and cell-type specificity (Oishi et al., 2017; Wu et al., 2023).

The retina is a highly energy-demanding tissue of the central nervous system (CNS) and is continuously exposed to fluctuating levels of light, oxygen and metabolic substrates (Joyal et al., 2018). Because ATP is rapidly turned over into adenosine diphosphate (ADP) and AMP, and ultimately adenosine, metabolic demand directly influences intracellular adenosine dynamics (Garcia-Gil et al., 2021). Moreover, changes in intracellular adenosine during the circadian cycle have been shown to govern extracellular adenosine concentrations via ENTs (Cao et al., 2020; Ribelayga and Mangel, 2005). Therefore, in the adult retina, adenosine levels fluctuate in response to daily physiological cues.

As a signaling molecule in the retina, adenosine is well characterized for its role in pathological angiogenesis such as that observed in the oxygen-induced retinopathy (OIR) model; adenosine acting via A2AR amplifies HIF-1α-dependent angiogenesis (Liu et al., 2017). Moreover, adenosine modulates stage I and II retinal waves to refine visual maps in retinofugal targets before eye opening (Arroyo and Feller, 2016; Huang et al., 2014; Syed et al., 2004; Torborg and Feller, 2005). Recent studies have demonstrated that multiple neurotransmitters involved in retinal waves (Biswas et al., 2020, 2024; Liang et al., 2023; Weiner et al., 2019) and light-dependent neural activity also influence retinal vascular development (D'Souza and Lang, 2020; Nguyen et al., 2019; Rao et al., 2013).

From development through adulthood, intra- and extracellular adenosine metabolism thus plays a pivotal role in the retina. Notably, CD73 is abundantly expressed in the rod-photoreceptor lineage (Koso et al., 2009) and modulates extracellular adenosine levels in the OIR model during retinal development (Zhang et al., 2022) and across adult light–dark cycles (Cao et al., 2020; Ribelayga and Mangel, 2005). However, CD73 and adenosine receptors are expressed in many retinal cell types, and their roles in retinal diseases such as diabetic retinopathy and age-related macular degeneration are complex (Fan et al., 2023; Santiago et al., 2020). Because this profile remains poorly defined, a detailed characterization of the CD73 profile (its expression and function as a trigger of adenosine metabolism) in the normal retina is essential for understanding adenosine biology in physiological and pathological contexts.

Here, we mapped the CD73 profile in the retina from embryogenesis through adulthood using newly generated transgenic mouse lines (*CD73-BAC-EGFP* and *CD73-CreER^{T2}*) and interpreted these findings by re-analyzing public single-cell RNA sequencing (scRNA-seq) datasets. We revealed developmentally regulated patterns of CD73 expression not only in the rod lineage but also in

the inner nuclear layer (INL), and transiently expressed in the early astrocyte lineage. CD73 deficiency altered rod photoresponses under dim light and caused time- and region-specific changes in vascular morphology. The intrinsic fluctuations, which would have been undetectable by receptor manipulation alone, lay the foundation for investigating how genetic and environmental risk factors disturb adenosine metabolism in retinal pathogenesis.

## RESULTS

### *CD73-BAC-EGFP* mice reveal that CD73 expression is primarily restricted to the rod lineage and robustly detectable from P3 onward

Previous studies have shown that CD73 is expressed in rods (Koso et al., 2009) and used as a rod-specific marker (Sarin et al., 2018). However, several reports have also noted its presence in Müller glia and the retinal pigment epithelium (RPE) (Chen et al., 2014; Wurm et al., 2011) (Fig. 1A). To resolve this discrepancy, we first examined CD73 expression in our *CD73-BAC-EGFP* (*CD73-EGFP*) mice (Breitbach et al., 2018) under physiological conditions. In adult retinas, EGFP signals were predominantly localized to the outer nuclear layer (ONL), corresponding to photoreceptors, and were undetectable in horizontal, bipolar, amacrine or Müller glial cells within the INL (Fig. 1B). Immunostaining with cone-arrestin (Arr3), a marker of cone photoreceptors (cones), confirmed that EGFP was absent from cones (Fig. 1C), indicating that rods are the primary source of EGFP signals in the ONL.

We next investigated CD73 expression during development. Around P3, the neuroblastic layer (NBL) splits into the outer neuroblastic layer (oNBL) and the inner neuroblastic layer (iNBL) that later form the ONL and the INL (Burger et al., 2021) (Fig. S1). CD73-EGFP expression became robust within the NBL from P3 onward (Fig. 1D). Although we observed only minimal EGFP expression in the NBL at P0–P1 in the whole-mount view (Fig. S2), we could not fully specify the retinal cell types prior to P3. Moreover, Arr3

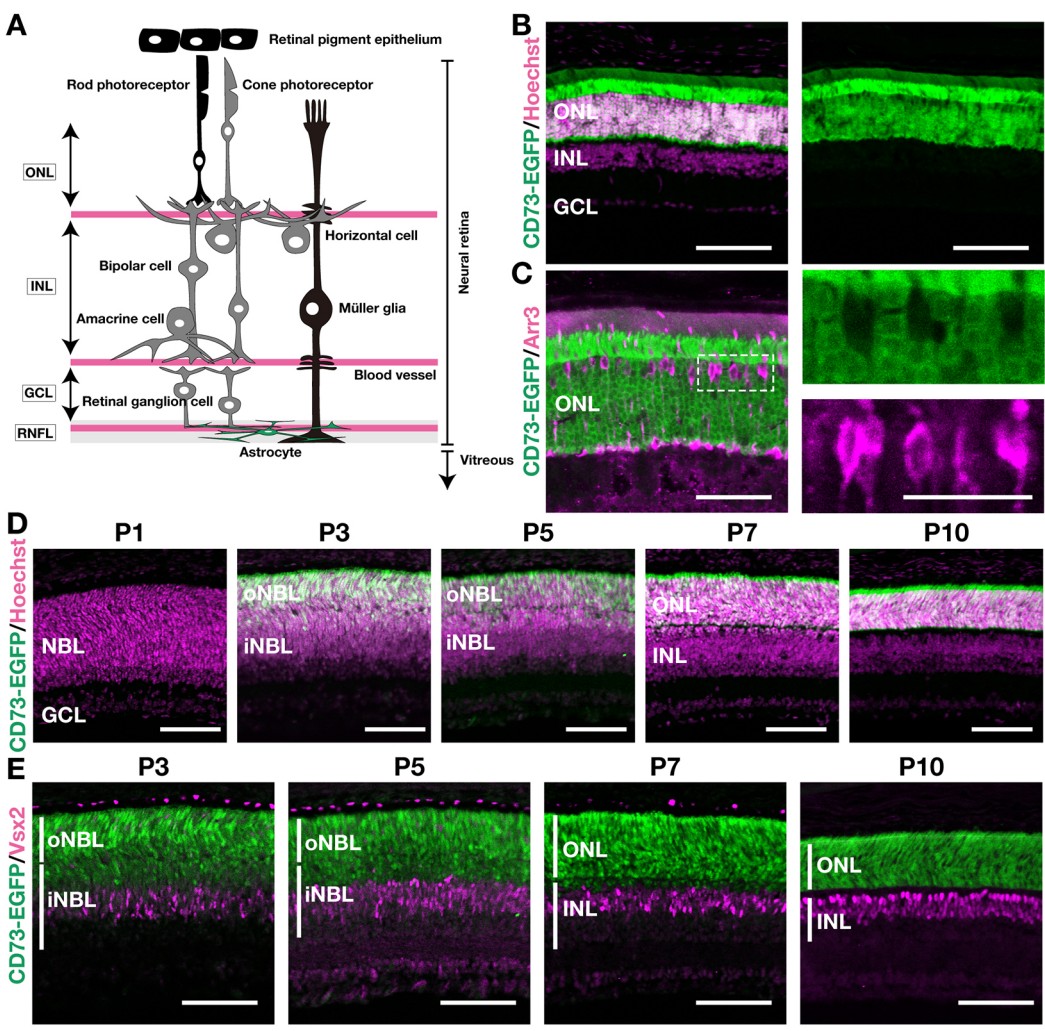

**Fig. 1. The *CD73-EGFP* mice reveal that CD73 is confined to the rod lineage and becomes detectable from P3 onward.** (A) Schematic illustration of adult retinal anatomy. Cell types drawn in black (RPE, rod photoreceptors and Müller glia) have been previously reported to express CD73.
(B) Representative immunofluorescence images of retinas from adult (2- to 3-month-old) *CD73-BAC-EGFP* mice. Left: merged image (CD73-EGFP in green; Hoechst in magenta). Right: single-channel image of CD73-EGFP. (C) Higher-magnification views. Left: merged image of Arr3 (cone arrestin, magenta) and CD73-EGFP (green). Right: higher-magnification view of the area outlined in the left panel; CD73-EGFP (green) channel (top), Arr3 (magenta) channel (bottom). (D) Developmental time course (P1-P10) of mouse retinas. EGFP (green) and Hoechst (magenta). (E) Immunostaining for the retinal progenitor and bipolar marker Vsx2 (magenta) in retinas from CD73-EGFP (green) mice. Scale bars: 100 μm (B,D,E); 50 μm (C, left); 25 μm (C, right). GCL, ganglion cell layer; iNBL, inner neuroblastic layer; INL, inner nuclear layer; NBL, neuroblastic layer; oNBL, outer neuroblastic layer; ONL, outer nuclear layer; RNFL, retinal nerve fiber layer; RPE, retinal pigment epithelium.

immunostaining confirmed that these EGFP[+] cells were primarily restricted to the Arr3-negative rod lineage rather than to cones (Fig. S3). Although EGFP was undetectable in the INL in adulthood, from P3 to P7, we occasionally detected EGFP signals in both the iNBL and oNBL (Fig. 1D). Because the rod lineage originates from the same late progenitor pool as bipolar cells (Brzezinski and Reh, 2015; Wang et al., 2014), we asked whether these transient signals might include bipolar precursors. Immunostaining for Vsx2 (Chx10), a marker of retinal progenitor cells and bipolar cells, showed that Vsx2-positive cells were EGFP negative (Fig. 1E), indicating that CD73-EGFP is not expressed in the bipolar lineage. These occasional EGFP[+] cells in the iNBL are therefore most likely rod precursors, consistent with the transient presence of rod lineage cells in this layer during early postnatal development (Burger et al., 2021). Thus, *CD73-EGFP* mice show that CD73 expression became detectable after P3 in the rod lineage and remains rod specific in adulthood under physiological conditions.

## *CD73-CreER^T2;tdTomato* mice further validate that predominant CD73 expression in the neural retina begins in the rod lineage from P3 onward

In CD73-EGFP mice, we also observed faint EGFP signals in the Pax2[+] astrocytes at early postnatal stages and rare EGFP[+] cells in the INL in the adult retinas (Fig. S4). To confirm these CD73-EGFP observations across multiple developmental stages, we generated another transgenic mouse line by crossing the *CD73-CreER^T2* knock-in mice with Rosa26-loxp-STOP-loxp-tdTomato (tdTomato) knock-in mice to obtain *CD73-CreER^T2; tdTomato* mice (Fig. 2A). We then performed the lineage tracing to determine how *CD73CreER^{T2+}* cells at tamoxifen-treated time points differentiate into various cell types at the time of sampling. First, to check for CreER^T2 leakiness, we examined mice that did not receive tamoxifen and confirmed that no tdTomato signal was detected (Fig. S5A). Next, to assess the adult expression of *CD73-CreER^T2*, we administered tamoxifen to 2- to 3-month-old mice 2 weeks prior to tissue collection (Fig. 2B).

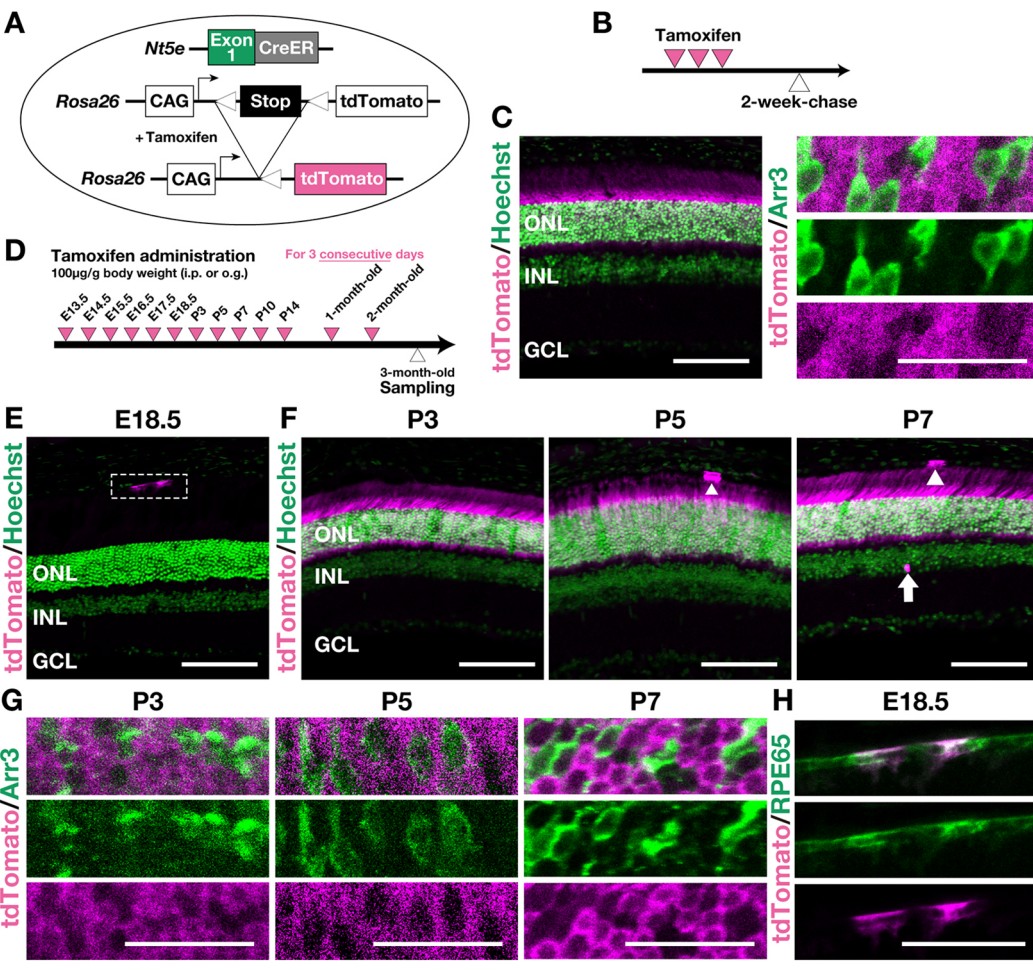

**Fig. 2. Lineage-tracing experiments confirm that CD73 expression in the ONL is restricted to the rod lineage from P3 onward.** (A) Schematic of the tamoxifen-inducible *CreER^T2* system for labelling CD73-fated cells. (B) Timeline showing tamoxifen injections (magenta arrowheads) and sample collection (white arrowhead) in adult mice. (C) Representative images of retinas from 3-month-old *CD73-CreER^T2;tdTomato* mice 2 weeks after tamoxifen injection. Left: Hoechst (green). Right: higher-magnification view of the outer region of the ONL. Arr3 (cone arrestin, green) marks cone photoreceptors. Top, merged; middle, Arr3; bottom, tdTomato alone. tdTomato is shown in magenta. (D) Schematic of the lineage-tracing strategy to visualize descendants of *CD73-CreER^T2+* cells from E13.5 to 2 months of age, with tissue collection at 3 months of age. In B and D, magenta arrowheads indicate tamoxifen injection days for each of the experimental groups; white arrowheads mark tissue collection. (E-H) Lineage tracing in *CD73-CreER^T2;tdTomato* mice treated with tamoxifen at E18.5, P3, P5 or P7 and analyzed at 3 months of age. (G,H) Higher-magnification views of F and E, respectively. In E, the white dashed boxes outline the regions shown in H. In E and F, Hoechst (green) labels nuclei. In G, Arr3 (green) marks cone photoreceptors: top, merged; middle, Arr3; bottom, tdTomato. In H, RPE65 (green) marks the RPE: top, merged; middle, RPE65; bottom, tdTomato. tdTomato[+] cells are absent in the neural retina at E18.5 (E), but they are predominantly located in the ONL at P3, P5 and P7 (F). The dashed outline in E indicates tdTomato[+] cells in the RPE, illustrated at higher magnification in H. Scale bars: 100 μm (C, left; E; F); 50 μm (H); 25 μm (C, right; G).

Previous work using *CD73-EGFP* mice showed CD73-EGFP expression in the endosteal tissue (Breitbach et al., 2018). To validate the Cre line, we confirmed that *CD73-CreER^T2;tdTomato* mice had tdTomato-labelled cells in the same endosteal region, consistent with *CD73-EGFP* mice result (Fig. S5B; arrowheads). In the retina, tdTomato-labelled cells were abundant in the ONL and Arr3-negative rods, in agreement with our CD73-EGFP results (Fig. 2C). Furthermore, in the *CD73-CreER^T2; tdTomato* mice, we also observed a small population of tdTomato⁺ cells in RPE65⁺ RPE (Fig. S6A), which was not clearly detected in the *CD73-EGFP* mice. This discrepancy may reflect the stronger tdTomato signal in *CD73-CreER^T2;tdTomato* mice compared with the faint CD73-EGFP signal shown in immature astrocytes and the INL (Fig. S4). Thus, we re-analyzed a public scRNA-seq dataset (Li et al., 2024) and confirmed *Nt5e* (CD73) expression in the RPE (Fig. S6B).

We then investigated earlier developmental stages by performing lineage-tracing experiments (Fig. 2D). When tamoxifen was given at E18.5, the tdTomato signal was not apparent in the ONL (Fig. 2E). However, tdTomato⁺ cells were clearly visible in the ONL when tamoxifen was administered after P3 (Fig. 2F). In contrast, few tdTomato⁺ cells appeared in the INL in any tamoxifen-treated group (Fig. 2F; Fig. S6C). Furthermore, we confirmed that these tdTomato⁺ cells in the ONL were Arr3 negative (Fig. 2G; Fig. S6D), indicating that the *CD73-CreER^T2* is expressed in the rod lineage. Moreover, previous reports have suggested that CD73 expression is mainly induced by HIF-1α in various tissues (Alcedo et al., 2021; Synnestvedt et al., 2002). Therefore, we examined *Nt5e* (CD73) expression along pesudotime in the rod lineage using a public scRNA-seq dataset. We found that *Nt5e* expression tended to correlate with the HIF-1 signature score (Fig. S7).

Although we could not fully characterize their distribution from transverse sections alone, we detected tdTomato⁺ RPE in tamoxifen-treated groups from embryonic stages onward (Fig. 2E,F and H, Figs S4C, S6C and S8A; arrowheads), suggesting that CD73 expression in the RPE is regulated by mechanisms distinct from those operating in the neural retina. Collectively, these lineage-tracing data reinforce our CD73-EGFP findings, demonstrating that CD73 expression in the neural retina is readily detectable in the rod lineage by P3 and persists into adulthood.

## CD73 is transiently expressed in the early astrocyte lineage from E16.5 to P3, and becomes apparent in the INL at P7, persisting into adulthood

In transverse sections, a small number of tdTomato⁺ cells were also observed in the INL (Fig. 2F, Fig. S6C; arrow) and in the retinal nerve fiber layer (RNFL), the thin layer immediately vitreal (inner) to the ganglion cell layer (GCL) (Fig. S8B; arrowheads; see Fig. 1A for an anatomical schematic). To further clarify the spatiotemporal distribution of these tdTomato⁺ cells in the INL and RNFL, we examined whole-mount retinas from *CD73-CreER^T2;tdTomato* mice. We found that tdTomato⁺/S100β⁺ (an astrocyte marker) cells in the RNFL were present during embryonic and early postnatal stages (Fig. 3A), whereas a few tdTomato⁺ cells in older retinas appeared in the INL (Fig. 3B). Most tdTomato⁺ somata in the INL were round and lay on the side facing the GCL (Fig. 3B, arrowheads). However, a few tdTomato⁺ cells exhibited distinct shapes that spanned from the ONL to the GCL (Fig. 3B, right, arrow; Fig. 3C, arrow). To better interpret the identities of tdTomato⁺ cells in the INL, we re-analyzed a public adult single-cell RNA-seq dataset (Li et al., 2024). *Nt5e* transcripts were detected in all major INL classes, including amacrine cells, rod

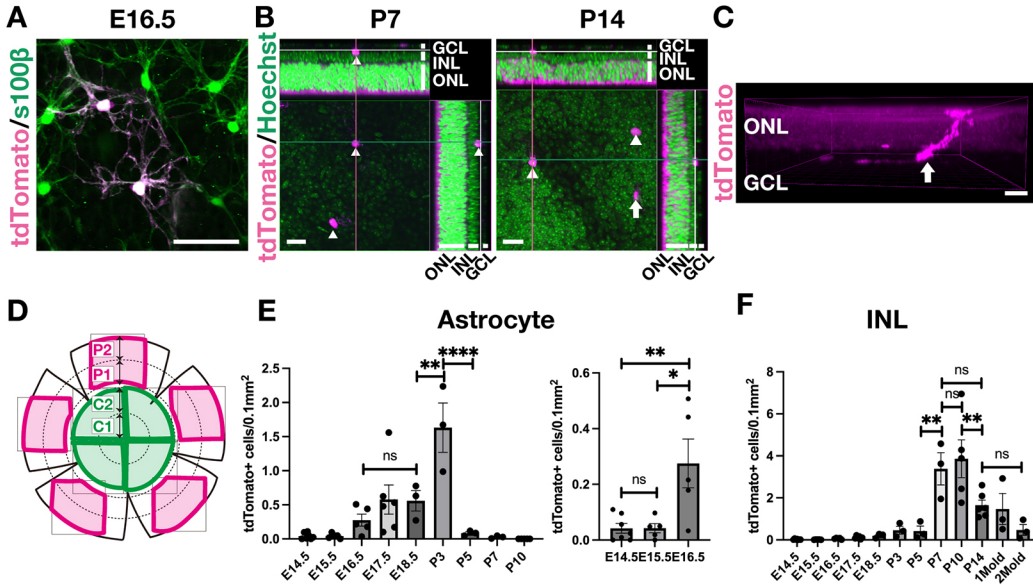

**Fig. 3. Temporal distribution of descendants of *CD73-CreER^T2+* cells in the RNFL and INL at 3 months of age.** (A) Representative image of tdTomato⁺ astrocytes in the RNFL. Tamoxifen was administered at E16.5. S100β (an astrocyte marker) in green; tdTomato in magenta. (B) Orthogonal views of representative tdTomato⁺ cells in the INL. Tamoxifen was administered at P7 (left) or P14 (right). The main panel shows the *xy* plane; the *xz* (top) and *yz* (right) planes are orthogonal slices (green and magenta lines, respectively; white lines in *xz* and *yz* indicate the *z*-position of the *xy* plane). Hoechst (green); tdTomato (magenta). Arrowheads indicate tdTomato⁺ cells located on the GCL-facing side of the INL. (C) 3D Imaris reconstructions of tdTomato⁺ cells at P14. The tdTomato⁺ cell indicated by the arrow corresponds to the one marked by the arrow in B (right). (D) Schematic illustrating tdTomato⁺ cell quantification. Whole-mount retinas were divided into central (green) and peripheral (magenta) regions. The central region was split into C1 (0-500 μm from the optic nerve head) and C2 (500-1000 μm), while the peripheral region was split into P1 (500-1000 μm from the periphery) and P2 (0-500 μm). tdTomato⁺ cells were counted in each zone. (E,F) Temporal characteristics of tdTomato⁺ cells in astrocytes (E) and the INL (F) in different tamoxifen-treatment groups (E14.5 to P10 in E, left; E14.5 to E16.5 in E, right; E14.5 to 2 months of age in F). Data are presented as mean±s.e.m., with each dot representing data from an individual sample (*n*≥3 retinas per condition, one retina per mouse). One-way ANOVA was used to compare tdTomato⁺ cell densities among time points, followed by Tukey's multiple comparisons (*P<0.05; **P<0.01; ****P<0.0001). All samples were collected at 3 months of age. Scale bars: 50 μm (A,B); 20 μm (C).

and cone bipolar cells, horizontal cells, and Müller glia (Figs S6B and S9A). Furthermore, subtype-level analyses of cone bipolar cells (Fig. S9B-D) and amacrine cells (Fig. S10) showed that *Nt5e* expression was broadly distributed across subtypes and was not restricted to any specific subtype. Together, our lineage-tracing data and scRNA-seq re-analysis indicate that CD73$^+$ cells in the INL are not confined to a single cell type or subtype.

To investigate these patterns in the astrocyte lineage and the INL, we performed quantitative analyses of the entire retina from the optic nerve head to the periphery (Fig. 3D; see Fig. S11 for anatomical details). First, we examined the temporal changes in descendants of CD73$^+$ cells in the astrocyte (tdTomato$^+$/S100β$^+$ cells). The number of tdTomato$^+$ cells increased until P3, then sharply declined by P5 and P7; by P10 no cells were detected (Fig. 3E, left). Although the density of tdTomato$^+$ cells differed markedly between E18.5 and P3, tamoxifen administration differed between these stages (oral gavage to the pregnant dam at E18.5 versus to the pups at P3), making a direct comparison unreliable. Although a few tdTomato$^+$ cells were present

at E14.5 and E15.5, their density increased substantially from E16.5 onward (Fig. 3E, right). By contrast, the number of lineage-traced descendants of CD73$^+$ cells (tdTomato$^+$ cells) in the INL increased gradually until P5, then surged at P7, and persisted into adulthood with only a slight decline thereafter (Fig. 3F).

We next investigated the spatial distribution of tdTomato$^+$ (descendants of *CD73-CreER$^{T2+}$*) astrocytes in 3-month-old *CD73-CreER$^{T2}$;tdTomato* mice. Tamoxifen was administered at E16.5, E17.5, E18.5 or P3, which correspond to the peak density of tdTomato$^+$ astrocytes (Fig. 3E). When tamoxifen was given at E16.5, descendants of *CD73-CreER$^{T2+}$* cells migrated centrifugally to the periphery (P1 and P2; Fig. 4A,B) rather than remaining in the central regions (C1 and C2; Fig. S12), although no statistically significant differences were found among these regions (Fig. 4C). In contrast, tdTomato$^+$ descendants were detected at statistically significant levels in the most peripheral areas for mice treated at E17.5, E18.5 or P3 (Fig. 4 and Fig. S12). Finally, we examined tdTomato$^+$ cells in the INL starting at P7, when their number began to increase (Fig. 3F).

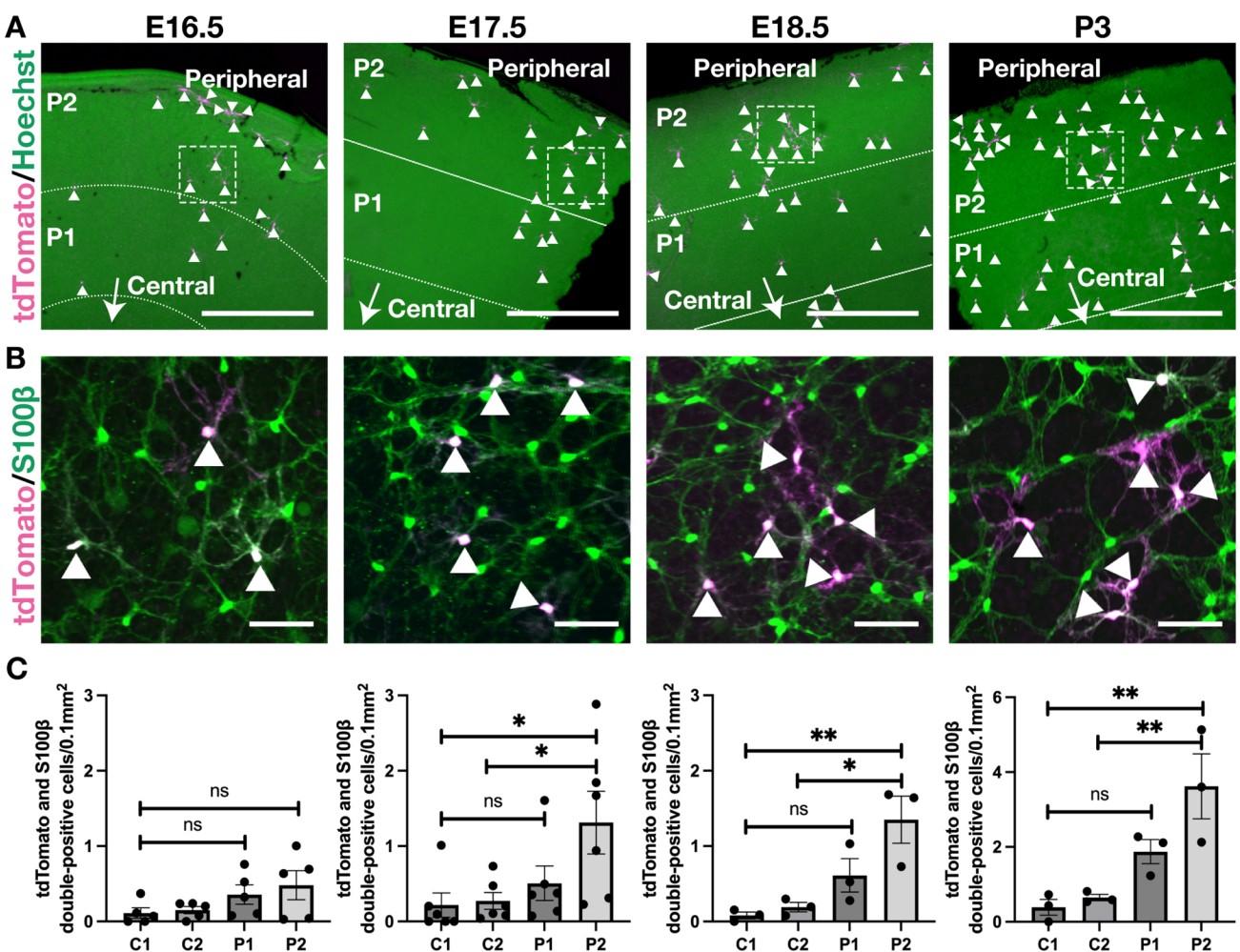

**Fig. 4. Descendants of *CD73-CreER$^{T2+}$* cells in the astrocyte lineage are enriched in the peripheral retina.** (A) Representative whole-mount retinal images from *CD73-CreER$^{T2}$;tdTomato* mice treated with tamoxifen at E16.5, E17.5, E18.5 or P3. Maximum-intensity projection images spanning from the INL to the RNFL, excluding the ONL, provide overviews of the peripheral retina for each tamoxifen treatment group. Hoechst (green); tdTomato (magenta). 'Peripheral' marks the retinal periphery; the arrow labeled 'Central' indicates the direction toward the optic nerve head. (B) Higher-magnification views of the areas outlined in A. S100β (an astrocyte marker) is in green; tdTomato is in magenta. Arrowheads indicate tdTomato and S100β double-positive cells. (C) The spatial distribution profiles of tdTomato$^+$ astrocytes, measured from the optic nerve head (0 µm) to the peripheral retina (covering C1, C2, P1 and P2). Data are shown as mean±s.e.m., with each dot representing data from an individual sample (*n*=3 for E16.5, *n*=6 for E17.5, *n*=3 for E18.5, *n*=3 for P3). One-way ANOVA was used to compare the density of tdTomato/S100β$^+$ astrocytes across these regions, followed by Tukey's multiple comparisons (**P*<0.05; ***P*<0.01). All samples were collected at 3 months of age. Scale bars: 500 µm (A); 50 µm (B).

Although the tdTomato[+] cells in the INL appeared slightly more concentrated in the central region (Fig. 5A,B; Fig. S13), there were no significant differences among all regions (Fig. 5C). Taken together, our lineage-tracing data indicate that CD73 is expressed within the early astrocyte lineage and its descendants accumulate in the periphery, whereas CD73[+] cells in the INL are distributed throughout the retina with a slight central enrichment.

### CD73 as an initiator of extracellular adenosine metabolism

Previous work has shown that mRNA expression of adenosine receptors in the retina increases around P5 (Koso et al., 2009). Studies using A2AR-tdTomato mice have shown that A2AR is mainly expressed in retinal vessels at P9 and P12 during development and in adulthood (Zhong et al., 2021). Therefore, to investigate how CD73-derived adenosine could act on the vasculature, we first re-analyzed adenosine receptor expression in endothelial cells using public scRNA-seq datasets at P6 and P10 (Zarkada et al., 2021). Among the adenosine receptors, *Adora2a* (A2AR) was expressed in endothelial cells, especially in tip cells, at both ages (Fig. S14A,B).

In the adult retina, *Adora2a* was expressed not only in endothelial cells but also in pericytes (Fig. S14C).

Although a previous study reported no obvious abnormality of retinal vascular development in CD73-knockout mice at P3, P12 and P17 (Zhang et al., 2022), we generated *CD73-CreER[T2]* knock-in/knockout (CD73 KO) mice by mating *CD73-CreER[T2]* mice to homozygosity (Fig. S15), and examined vascular phenotypes in CD73 KO mice at P3, P5, P7, P10, P14 and 3 months of age to investigate retinal vascular development in more detail (Fig. S16; Fig. S17). Consistent with the previous report, we did not detect gross abnormalities at most stages (Fig. S16; Fig. S17). However, when we quantified vascular morphology by measuring Simple Neurite Tracer-based total cable length (Fig. S17D), we found a small increase in total cable length in the deep plexus at P10 and a small decrease in total cable length in the deep and intermediate plexuses at 3 months of age within the 500-1000 μm annulus (Fig. S17E,F). These results suggest that CD73 deficiency does not strongly disturb overall vascular patterning under physiological conditions, but may subtly influence vascular complexity in specific regions and time windows.

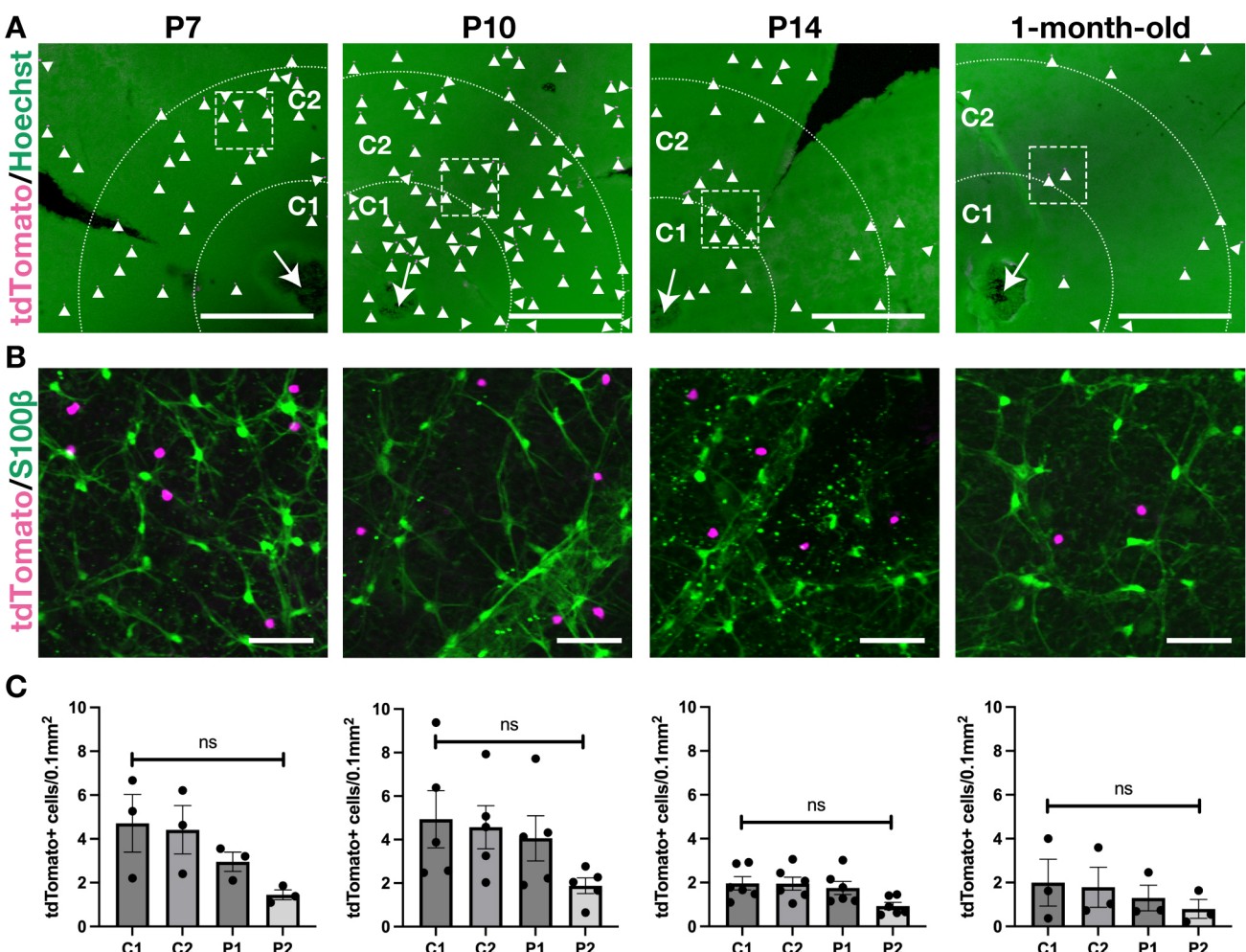

**Fig. 5. Spatial distribution patterns of descendants of *CD73-CreER[T2]* cells in the INL.** (A) Representative whole-mount retinal images from *CD73-CreER[T2]; tdTomato* mice treated with tamoxifen at P7, P10, P14 or 1 month of age. Maximum-intensity projection images are shown from the INL to the RNFL, excluding the ONL, providing an overview of the central retina for each treatment group. Hoechst (green); tdTomato (magenta). Arrowheads indicate tdTomato[+] cells in the INL. The arrow indicates the optic nerve head. (B) Higher-magnification views of the white dashed boxes in A. S100β (an astrocyte marker) is in green; tdTomato is in magenta. These images confirm that the tdTomato[+] cells in (A) reside in the INL and lack S100β expression. (C) The density of tdTomato[+] cells in the INL, measured from the optic nerve head (0 μm) to the peripheral retina. Data are shown as mean±s.e.m., with each dot representing data for an individual sample (*n*=3 for P7, *n*=5 for P10, *n*=6 for P14, *n*=3 for 1-month-old mice). One-way ANOVA was used to compare the density of tdTomato[+] cells in the INL across the four regions (C1, C2, P1 and P2), followed by Tukey's multiple comparisons. All samples were collected at 3 months of age. Scale bars: 500 μm (A); 50 μm (B).

Because purinergic signaling is closely linked to inflammatory status, we next examined adenosine-related genes in microglia (Fig. S18) and microglial distribution (Fig. S19). In adult single-cell RNA-seq data (Li et al., 2024), *P2ry12* transcripts (a purinergic ADP receptor widely used as a microglial marker) were higher than those of any adenosine receptor, and among the adenosine receptors *Adora3* showed the highest levels (Fig. S18A). We then confirmed P2Y12 protein expression in microglia by co-staining P2Y12 with Iba1 (Fig. S18B). Finally, we quantified Iba1$^+$ microglia in the OPL. Iba1$^+$ cells, particularly in the regions closer to the optic nerve head (500-1000 µm annulus during development and in the 0-1250 µm annulus at 3 months of age), tended to be slightly more numerous in CD73 KO mice than in wild-type mice, but these trends did not reach statistical significance at any time point (Fig. S19). Taken together, these vascular and microglial findings indicate that CD73 KO retinas remain grossly normal under physiological conditions, but that CD73-dependent adenosine metabolism may subtly modulate vascular and microglial homeostasis.

### CD73 is dispensable for retinal cell differentiation under physiological conditions

In addition to canonical receptor-dependent pathways, recent studies have shown that receptor-independent mechanisms that drive epigenetic modifications (Garcia-Gil et al., 2021; Xu et al., 2017). Therefore, we examined retinal cell differentiation by quantifying transverse sections at P14, as described previously (Zhang et al., 2023). Our analyses with *CD73-BAC-EGFP* and *CD73-CreER^T2;tdTomato* mice showed that CD73 becomes distinctly detectable in the rod lineage by P3 (Fig. 1D,E; Fig. S3; Fig. 2F,G; Fig. S6C,D; Fig. S8A), suggesting that late-born retinal cells (bipolar cells, Müller glia, and rods) might be influenced by CD73 activity. Because most tdTomato$^+$ somata in the INL faced the GCL and were considered to be amacrine cells (Fig. 3B), we also confirmed their identity. Immunostaining for Vsx2$^+$ retinal bipolar cells, Lhx2$^+$ Müller glia and AP2α$^+$ amacrine cells (Fig. S20A) revealed no statistically significant differences among these cell types between wild-type and CD73 KO retinas (Fig. S20B). Moreover, we evaluated rod development by measuring ONL thickness in Hematoxylin- and Eosin-stained (H&E) sections, which was comparable across genotypes (Fig. S20C). These findings indicate that, under physiological conditions, CD73 deficiency does not affect retinal cell differentiation.

### The impact of CD73 dysfunction on retinal photoresponses is prominent under scotopic conditions but becomes negligible under photopic conditions

In the adult retina, most extracellular adenosine under physiological conditions is generated via the CD73 pathway, and these levels change during light/dark adaptation (Ribelayga and Mangel, 2005). A recent study showed that intravitreal administration of a CD73 inhibitor in dark-adapted C57BL/6N mouse eyes had no effect on a-wave or b-wave amplitudes (Losenkova et al., 2022). However, whether the complete loss of adenosine production by CD73 induces physiological and morphological changes in the mouse retina under normal light conditions remains unclear. To investigate the impact of CD73-mediated adenosine on retinal function, we performed scotopic and photopic ERG analyses in wild-type and CD73 KO mice. In the scotopic ERG, the average waveforms evoked by various flash intensities are shown (Fig. 6A). At moderate or bright light intensities, photoresponses were comparable between wild-type and CD73 KO mice (Fig. 6A, middle and right). However, under the

dim light conditions, CD73 KO mice exhibited slightly accelerated response kinetics relative to wild-type mice (Fig. 6A, left). Notably, the amplitudes of the a-wave and b-wave, which primarily reflect rod photoreceptors and ON bipolar cells, respectively, were not significantly different between the two genotypes (Fig. 6B, left and middle). In contrast, the implicit time (time-to-peak) of the b-wave, particularly under dim light conditions, was modestly shorter in CD73 KO mice compared to wild-type mice (Fig. 6B, right). Additionally, the initial activation phase of the a-wave was virtually identical between wild-type and CD73 KO mice, suggesting no effect on the activation processes of the rod phototransduction cascade (Fig. S21). In the photopic ERG, where the rod responses are suppressed by background illumination and the waveforms primarily reflect cone function, the amplitudes and implicit times of photopic b-waves were similar between CD73 KO and wild-type mice (Fig. 6C,D). These findings indicate that CD73-derived adenosine activity, while negligible under bright light conditions, plays a more pronounced role under low-light conditions. Collectively, *in vivo* ERG analysis suggests that the adenosine produced by CD73, which is expressed in rod photoreceptors, as confirmed by the results of our *CD73-EGFP* and *CD73-CreER^T2;tdTomato* mice (Figs 1C and 2C), contributes more prominently to rod-driven retinal responses than to cone-driven responses.

### CD73 modulates rod recovery under low-light conditions

To directly examine the influence of CD73 on the physiological function of rod photoreceptors, we performed single-cell recordings from individual rods of CD73 KO and wild-type mice (Fig. 7A). The maximal response amplitude (R$_{max}$) and half-saturating response intensities (I$_{1/2}$), which indicates the sensitivity of rod, were comparable between wild-type and CD73 KO rods (Fig. 7B-E). Under dim-light conditions, while the time-to-peak of the dim flash responses was unaffected (Fig. 7F), the integration time was significantly prolonged in CD73 KO rods compared with wild-type rods (Fig. 7G). This extended integration time may explain the shorter implicit time of b-wave observed in ERG recordings under dim light conditions. Since the ERG a-wave, which originates from photoreceptors, is a negative-going (downward) response, its delayed recovery may prolong the suppressive influence on the subsequent positive-going (upward) b-wave after its peak. This could result in an apparently earlier peak of the b-wave, even if the bipolar kinetics are unchanged. The prolonged integration time observed in CD73 KO rods may reflect a delay in the recovery processes of the rod phototransduction cascade, potentially due to rhodopsin phosphorylation by GRK1 or to reduced the GTPase activity of the transducin-PDE complex.

Next, at P14 and 1 month of age, overall retinal morphology did not differ between wild-type and CD73 KO mice (Fig. S19C; dashed lines in Fig. S22A). At 3 months of age, ONL thickness at the region located 1000 µm inferior to the optic nerve head tended to be greater in CD73 KO mice (49.0±1.3 µm, *n*=9) than in wild-type mice (43.4±1.3 µm, *n*=10). Although this difference did not reach the statistical significance after correction for multiple comparisons, it was larger than at other regions (solid lines in Fig. S22A). This inferior region is known to receive the highest light exposure and shows light-induced degeneration in *Crb1* mutations (e.g. *Crb1$^{-/-}$* and *Crb1rd$^{8/rd8}$*) (Alves et al., 2014; Mehalow et al., 2003; van de Pavert et al., 2004, 2007), but we did not observe comparable ONL disorganization in CD73 KO mice (Fig. S22B). Taken together, our results indicate that CD73 deficiency led to delayed recovery of rod photoresponse and may subtly influence ONL thickness in the highest light exposure region.

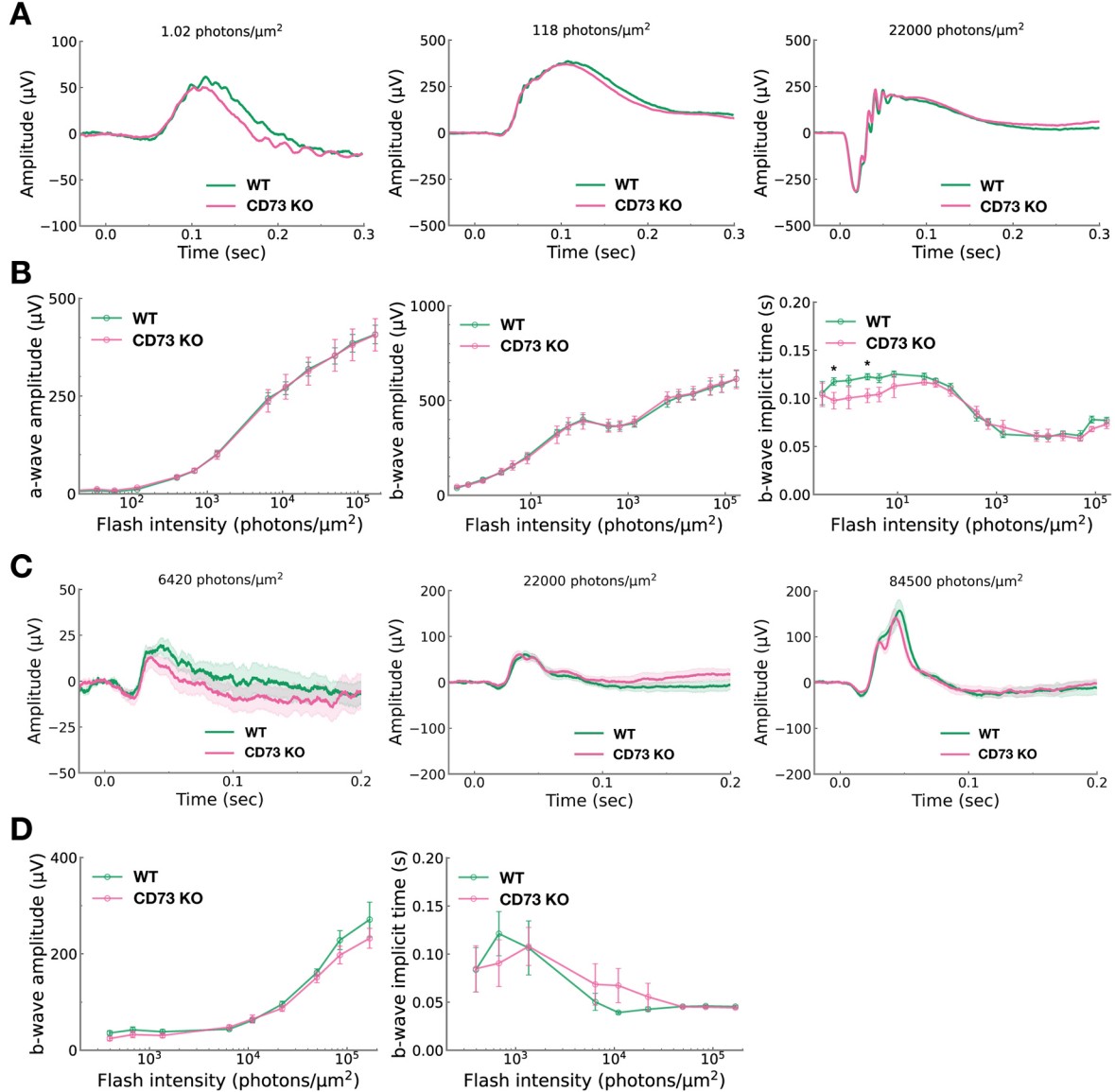

**Fig. 6. Scotopic and photopic ERG recordings in wild-type and CD73 KO mice.** (A) The average photoresponses recorded from wild-type (green line, $n$=10) and CD73 KO mice (magenta line, $n$=10) under scotopic conditions. The flash stimuli at time 0 were delivered with photon densities of 1.02 (left), 118 (middle), or 22,000 (right) photons/µm² on the cornea surface. (B) The plot of a-wave amplitude (left), b-wave amplitude (middle) and b-wave implicit time (right) in the scotopic ERG as a function of light intensity for wild-type (green, $n$=10) and CD73 KO (magenta, $n$=10) mice. Data are presented as mean ±s.e.m. (C) The average photoresponses recorded from wild-type (green line, $n$=9) and CD73 KO mice (magenta line, $n$=10) under photopic conditions. The intensity of flash stimuli on the cornea surface is 6420 (left), 22,000 (middle) and 84,500 (right) photons/µm². The shaded band around each mean response trace represents the s.e.m. (D) The plot of b-wave amplitude (left) and b-wave implicit time (right) in the photopic ERG as a function of light intensity for wild-type (green, $n$=9) and CD73 KO (magenta, $n$=10) mice. Data are shown as mean±s.e.m. Statistical significance was assessed using an unpaired two-tailed Student's $t$-test (*$P$<0.05). All ERG recordings were performed in 3-month-old mice. Abbreviations: KO, knockout; WT, wild type.

## DISCUSSION

In this study, we have identified two key characteristics of the CD73 profile. First, we delineated the spatiotemporal pattern of CD73 expression from embryogenesis through adulthood, demonstrating that CD73, in addition to its well-established presence in rods, is also expressed in the INL and transiently in the early astrocyte lineage. Second, we detected a diminished rod photoresponse following CD73 disruption (Fig. 8). Our current findings under physiological conditions raise the mechanistic hypothesis that CD73 expression and function are regulated by the balance between retinal waves, light and HIF-1/2 (Fig. S23; Biswas et al., 2020; D'Souza and Lang, 2020; Paisley and Kay, 2021; Tao and Zhang, 2014).

### Hypoxia as a potential upstream regulator of the CD73 profile in the retina

Neonatal retinal *Hif1a* mRNA levels rise during the early-postnatal window (Caprara et al., 2011), paralleling the onset of our CD73 expression in the rod lineage from P3 onward. Consistent with this, our HIF signature analysis using a public scRNA-seq dataset showed that *Nt5e* expression gradually increased and became enriched in rods along this trajectory, whereas HIF-2 signature scores declined and the HIF-bias score increased, indicating that HIF-1-mediated transcription becomes relatively predominant at later stages and is associated with *Nt5e* expression.

In the astrocyte lineage, our lineage-tracing results show that descendants of CD73⁺ astrocytes accumulate in the peripheral

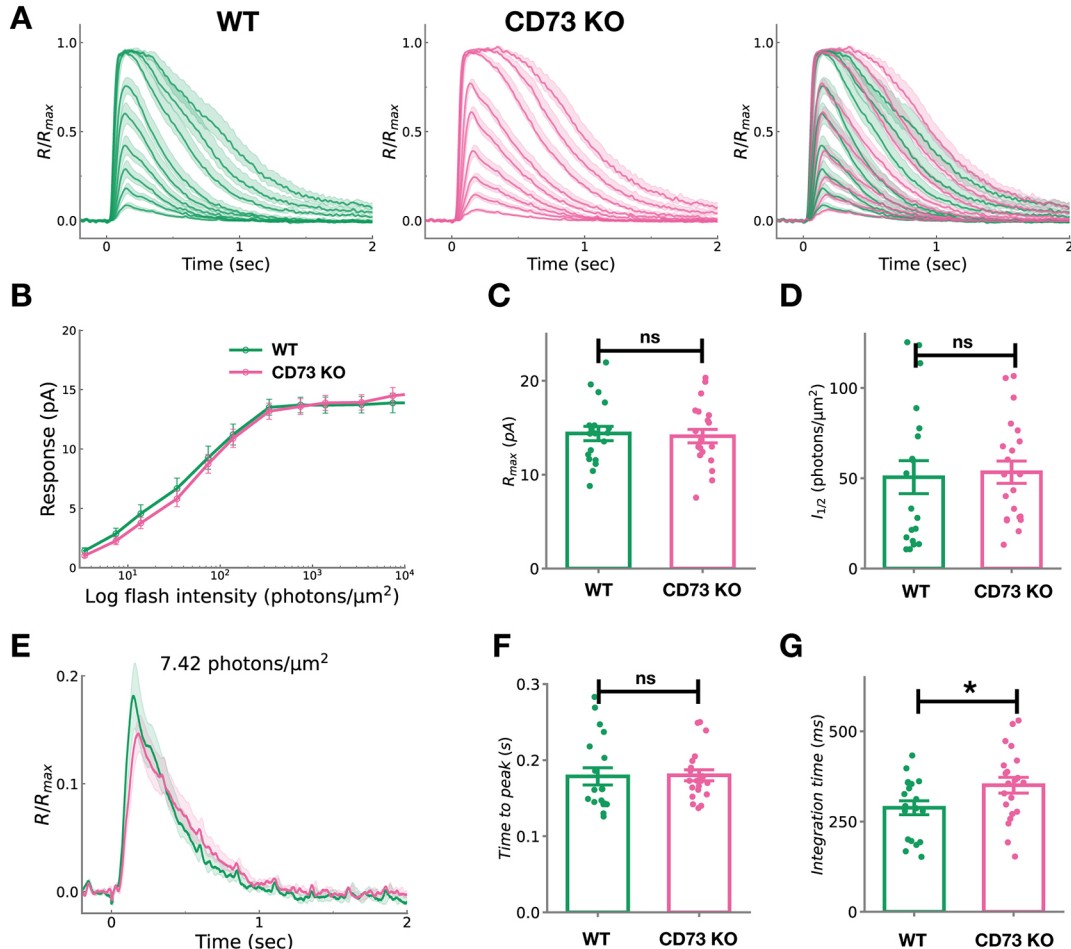

**Fig. 7. Single-cell recordings of wild-type and CD73-KO rod photoreceptors.** (A) A series of averaged responses of wild-type (left; $n$=19 rods) and CD73 KO (middle; $n$=21 rods) rod photoreceptors to flashes of varying intensities: 3.4, 7.42, 13.8, 34, 74.2, 138, 340, 742, 1380 and 3400 photons/μm². The right panel shows the merged responses of wild-type and CD73 KO rods. Responses from each rod photoreceptor were normalized to the maximal response amplitude of the respective photoreceptor. The shaded band around each mean response trace represents the s.e.m. (B-D) The intensity-response relationship in wild-type (green, $n$=19 rods) and CD73 KO (magenta, $n$=21 rods) rod photoreceptors (B). The maximal amplitude (C) of the response and $I_{1/2}$ (D) in wild-type and CD73 KO rod photoreceptors. Data are shown as mean±s.e.m. (E-G) Average response evoked by a flash (7.42 photons/μm²) of wild-type (green, $n$=19 rods) and CD73 KO (magenta, $n$=21 rods) rod photoreceptors (E). The time to peak (F) of the response and integration time (G) of the dim-flash response in wild type and CD73 KO. The shaded band around each mean response trace represents the s.e.m. (E). (F,G) Data are shown as mean ±s.e.m., with each dot representing data from an individual sample. Statistical significance was assessed using an unpaired two-tailed Student's $t$-test (*$P$<0.05). All single-cell recordings were performed in 3-month-old mice.

retina. Pax2⁺ astrocyte progenitor cells migrate centrifugally and gradually transition into immature astrocytes that begin to upregulate GFAP. These early astrocyte lineages reach the retinal margin by P3-P4 and remain mitotically active until P5-P6 (Chan-Ling et al., 2009; Duan et al., 2017, 2023; Tao and Zhang, 2014; West et al., 2005). These observations indicate that CD73 is transiently upregulated in the early astrocyte lineage during migration and downregulated once maturation is under way.

Although genetic studies using GFAP-Cre show that deleting HIF-2α does not disturb physiological vascular development (Weidemann et al., 2010), Pax2⁺/GFAP⁻ astrocyte-progenitor cells rely more on HIF-2α than on HIF-1α (Duan et al., 2014). HIF-2α becomes the dominant isoform under moderate or chronic hypoxia, whereas HIF-1α is preferentially stabilized during acute or severe hypoxia. Over time, cellular activation therefore shifts from HIF-1α to HIF-2α (Holmquist-Mengelbier et al., 2006; Koh and Powis, 2012; Sato and Yanagita, 2013; Semba et al., 2016). These previous reports, together with our re-analysis of public scRNA-seq data along the rod-lineage pseudotime, suggest that the balance

between HIF-1α and HIF-2α is also likely to be correlated with *Nt5e* expression in the astrocyte lineage.

### Light-dependent regulation of adenosine metabolism implied by temporal CD73 expression

Adenosine modulates spontaneous stage I (E16.5-P0/1) and stage II (P0/1–P10) retinal waves (Huang et al., 2014; Syed et al., 2004; Torborg and Feller, 2005). The onset of CD73 in the astrocyte lineage at E16.5 coincides with the stage I initiation, suggesting that extracellular CD73-generated adenosine could modulate these waves. By contrast, the later time points (P3, when CD73 expression in the astrocyte lineage declines, and P7, when it begins in the INL) do not align with either the onset or offset of retinal waves. These results raise the possibility that CD73 expression is not regulated simply by mechanisms related to retinal-wave activity during early postnatal development.

Recent studies have shown that the vitreous hyaloid vasculature undergoes light-dependent remodeling, mediated first by the OPN4–VEGFA pathway (E16-P3) and later by the OPN5–dopamine

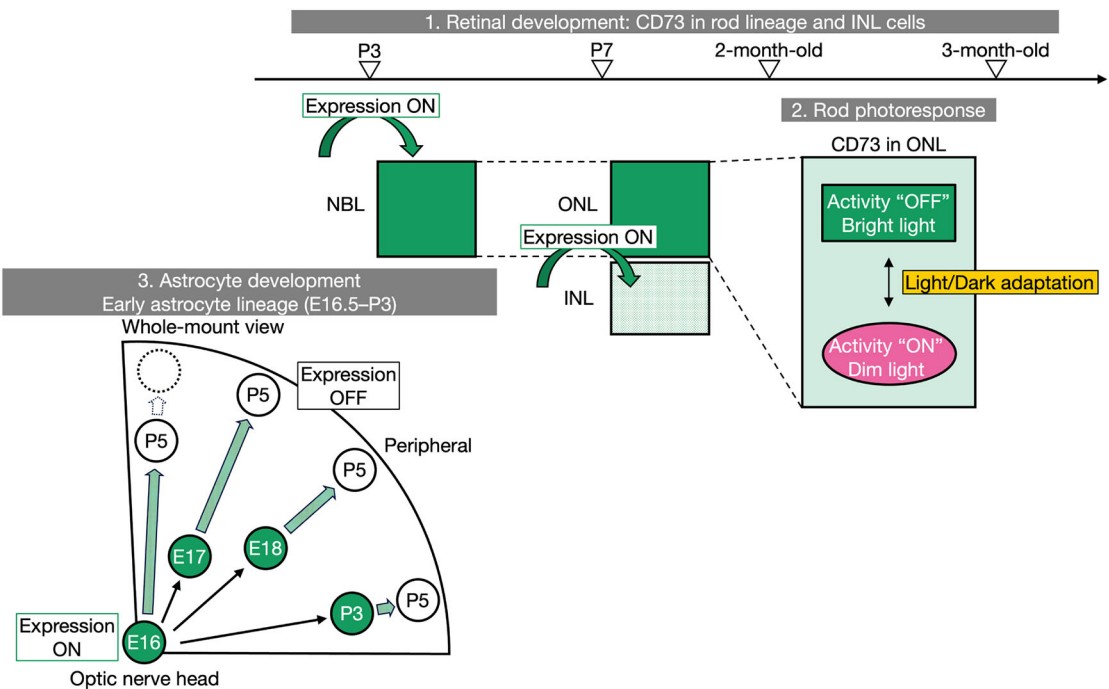

**Fig. 8. Proposed model of CD73 dynamics in the mouse neural retina under physiological conditions.** Spatiotemporal CD73 dynamics follows three patterns. (1) From P3 onward, it is abundant in the rod lineage; by P7 it is also initiated in a subpopulation of cells in the INL. CD73 expression in both rods and the INL persists into adulthood. (2) CD73 activity in rods becomes detectable ('ON') under dim-light conditions, whereas it remains undetectable ('OFF') under brighter conditions. (3) From E16.5 to P3, CD73 is expressed in a subpopulation of the astrocyte lineage that migrates all the way to the periphery, although its expression is rapidly lost after P3. The descendants of CD73+ cells at E16.5 do not show a statistically significantly accumulation in the adult retinal periphery.

pathway (P3-P8); OPN4 and OPN5 are expressed in distinct populations of RGCs (Nguyen et al., 2019; Rao et al., 2013). The pathway switch at P3 coincides with the loss of CD73 in the astrocyte lineage. Because dopamine and adenosine generally work in opposite directions (Cameron et al., 2020; Li et al., 2013), the loss of CD73 at P3 could be significant for the onset of dopamine release by dopaminergic amacrine cells. In addition, the initiation of phototransduction in photoreceptors around P8 (Bonezzi et al., 2018; Tiriac et al., 2018) coincides with the onset of CD73 expression in the INL at P7. From P7 onward, lineage tracing showed that CD73 was sparsely expressed in the INL. Although we still cannot determine whether this pattern reflects low-level but stable CD73 expression or a transient expression, these observations support the hypothesis that light stimulation, even before eye opening, contributes to the mechanisms that regulate CD73 expression.

Finally, we detected CD73 expression in the RPE from early development through adulthood, independent of the other expression patterns. During development, the RPE releases ATP into the extracellular space to modulate retinal progenitor proliferation (Pearson et al., 2005). In adulthood, the RPE both supports the high metabolic demands of photoreceptor phototransduction (Hurley, 2021) and phagocytoses rod outer segments in a circadian manner (Bhoi et al., 2023), and HIF-1α becomes active in response to changes in oxygen tension within the RPE (Forooghian et al., 2007; Kurihara et al., 2016). Outer retinal homeostasis may influence CD73 dynamics, thereby modulating the intra- and extracellular ATP/adenosine metabolism.

In summary, our spatiotemporal analysis across retinal progenitor and astrocyte differentiation supports the hypothesis that gradients of hypoxia, retinal-wave activity, light and adenosine metabolism may jointly regulate CD73 expression as the retina passes through

key developmental milestones. Future interventional studies, such as manipulating light exposure or oxygen conditions, should test this mechanistic hypothesis for the regulation of CD73 expression from retinal development through adulthood.

### Loss of CD73 disrupts normal recovery of rod photoresponses

In retinal neurons, A2AR are predominantly expressed in the inner retina (Chen et al., 2024; Santiago et al., 2020; Zhong et al., 2021). Although *in situ* hybridization likewise shows that *Adora2a* (A2AR) mRNA is mainly expressed in the inner retina with only a weak signal detected in the ONL (Kvanta et al., 1997), its expression has been reported to rise in cones during the subjective night (Li et al., 2013).

Electrophysiologically, extracellular adenosine acts in two opposite ways via A2AR. In dark-adapted rods, extracellular adenosine binds to A2A-like receptors on the synaptic terminal, lowers L-type $Ca^{2+}$ influx and thereby reduces $Ca^{2+}$-dependent glutamate release; pharmacological blockade of these receptors does not alter the photovoltage of the rod (Stella et al., 2003). Conversely, when adenosine acts on A2AR expressed by cones, extracellular adenosine increases rod–cone gap-junction coupling, thereby strengthening rod-driven cone responses and enhancing night-time cone sensitivity (Cao et al., 2020; Li et al., 2013).

By contrast, our CD73 KO mice are presumably exposed to chronically low extracellular adenosine during the subjective day. In photopic ERG, the CD73 KO showed no deficit in cone-driven responses. In contrast to the earlier pharmacological receptor blockade data, our single-cell recordings for rods revealed that genetic deletion of CD73 delayed response recovery in phototransduction.

We did not directly measure retinal adenosine levels in the present study. However, previous work has shown that extracellular adenosine levels are shaped by ENT-dependent bidirectional adenosine flux and that this flux favors extracellular-to-intracellular uptake during the subjective day (Cao et al., 2020; Ribelayga and Mangel, 2005). Therefore, we hypothesized that CD73 deficiency might reduce extracellular adenosine pools and compromise adenosine salvage, thereby limiting ATP availability in rods. Real-time ATP imaging in rods should clarify these alternative mechanisms (He et al., 2021).

### CD73 function in the retinal adenosine metabolism from development through adulthood

As in the adult retina, an OIR model study showed that both CD73-mediated and bidirectional ENT-mediated pathways can also act as sources of extracellular adenosine during development (Zhang et al., 2022). These previous reports suggest that bi-directional flux through ENTs compensates for CD73 deficiency. Although previous studies have shown that genetic inactivation of A1R, A2AR or CD73 does not cause obvious abnormalities during normal vascular development (Liu et al., 2010; Zhang et al., 2015, 2022), we examined vascular development with temporal and spatial resolution, and found that the total cable length in the deep plexus within the 500–1000 µm annulus from the optic nerve head at P10 was specifically increased in CD73 KO retinas compared with wild type. Our re-analysis of scRNA-seq data showed that endothelial tip cells highly express *Adora2a*, and these results support the hypothesis that adenosine metabolism is subtly disturbed during this developmental stage.

Moreover, although there was no statistical difference, CD73 KO mice tended to show the greatest ONL thickness ∼1000 µm inferior to the optic nerve head compared with wild type. Adult retinal thickness can vary as part of emmetropization, but the spatial bias of this trend is noteworthy. Light exposure preferentially upregulates Müller-glia-derived growth factors such as bFGF in the inferior retina (Stone et al., 1999), and CD73 deficiency impairs Müller-glia volume regulation under hypo-osmotic stress (Wurm et al., 2010). In addition, at 3 months of age, the total cable length in the deep and intermediate plexuses in the inner region (500-1000 µm) was reduced in CD73 KO mice. Our re-analysis of public scRNA-seq data showed *Adora2a* expression not only in endothelial cells but also in pericytes. Taken together, these observations support the idea that loss of CD73 perturbs overall retinal homeostasis. These findings are hypothesis generating and may help to elucidate the pathology of diabetic retinopathy and age-related macular degeneration (Fan et al., 2023; Santiago et al., 2020).

In conclusion, our study reveals that CD73 expression undergoes a spatiotemporal shift during development, and that CD73 in rods contributes to the proper termination of phototransduction during adulthood. Dissecting how everyday physiological cues (e.g. hypoxia, light exposure, energy metabolism and circadian rhythms) modulate CD73 will help disentangle redundant pathways of adenosine metabolism and inform future studies of retinal disease.

## MATERIALS AND METHODS
### Mice

The *CD73-BAC-EGFP* (*CD73-EGFP*) mouse line has been described previously and was generously provided by Dr Bernd K. Fleischmann (University of Bonn, Germany; Breitbach et al., 2018). To generate the *CD73-CreER^{T2}* knock-in mouse line (*Nt5e*^{CreERT2}), a CreERT2 cassette was inserted into exon 1 of the mouse Nt5e locus using a CRISPR/Cas9-based genome editing platform, resulting in heterozygous expression of CD73. *CD73-CreER^{T2}* knock-in/knockout (CD73 KO) mice were generated by crossing *CD73-CreER^{T2}* knock-in mice as homozygotes. For lineage-tracing experiments, *CD73-CreER^{T2}* mice were crossed with Rosa26-loxp-

STOP-loxp-tdTomato (tdTomato) knock-in mice (The Jackson Laboratory, stock 007909; RRID: IMSR_JAX:007909). Wild-type C57BL/6J mice (The Jackson Laboratory, stock 000664; RRID: IMSR_JAX:000664) were purchased from CLEA Japan. All mice were bred and maintained under a 12 h light/dark cycle [lights on at 7:00 A.M. (ZT0), lights off at 7:00 P.M. (ZT12)] and used for experiments between ZT3 (10:00 A.M.) and ZT9 (4:00 P.M.). Both male and female mice were used, and data from both sexes were pooled because no sex-specific differences were detected. All animal procedures were approved by the Animal Care and Use Committee of the University of Tsukuba (Approval No. 23-432) and complied with institutional and governmental guidelines. Mice were euthanized by intraperitoneal injection of a triple anesthetic mixture (8 mg/kg midazolam, 1.5 mg/kg medetomidine and 10 mg/kg butorphanol tartrate, or higher if required), followed by cervical dislocation to ensure complete euthanasia.

### Mouse genotyping

Genomic DNA was extracted from mouse tail biopsies and used as a template for PCR-based genotyping. Two sets of primers were employed to detect the presence of either the *CD73-CreER^{T2}* knock-in allele or the wild-type CD73 allele. Primer pairs (Thermo Fisher Scientific; this study) were: *CD73-CreER^{T2}* knock-in (fw, 5′-CAGGCTCTAGCGTTCGAACG-3′; rev, 5′-ATTCTCCCACCGTCAGTACG-3′; product size, 182 bp) and CD73 WT (fw, 5′-AGTTTAGTAGAGGCCCCGGT-3′; rev, 5′-CACTTGGTGGAGTCATCGCT-3′; product size, 235 bp).

### Tamoxifen treatment

For lineage tracing, 1- to 2-month-old *CD73-CreER^{T2}* mice (both sexes) received tamoxifen (2 mg per mouse per day; Sigma-Aldrich; 20 mg/ml in corn oil) via intraperitoneal injection for 3 consecutive days. Neonatal mice (≤14 days old) received a single oral dose of tamoxifen (100 µg/g body weight). For embryonic-stage analyses, pregnant dams received a single 2 mg oral dose of tamoxifen. All tamoxifen-treated mice were raised to 3 months of age before euthanasia. To assess potential Cre 'leakiness', we also analyzed *CD73-CreER^{T2};Rosa-tdTomato* mice that did not receive tamoxifen.

### Staining of retinal sections

Enucleated eyes were fixed in 4% (w/v) paraformaldehyde (PFA) for 45 min at room temperature, then rinsed in PBS. Fixed tissues were cryoprotected overnight in 30% (w/v) sucrose in PBS at 4°C, embedded in OCT compound (Tissue-Tek, Sakura) and rapidly frozen. Retinal sections (12 µm) were cut on a cryostat and air-dried before staining.

For Hematoxylin and Eosin (H&E) staining, sections were briefly rinsed in distilled water, stained with Hematoxylin (Wako, 131-09665) for 5 min, and counterstained with eosin Y (Wako, 058-00062) for 45 s. After dehydration in a graded ethanol series, sections were mounted with Entellan (Merck Millipore).

For immunostaining, sections were blocked in PBS containing 3% donkey serum and 0.3% Triton X-100 for 1 h at room temperature. Primary antibodies, diluted in the same blocking solution, were applied overnight at 4°C. After washing in PBS, sections were incubated for 1 h at room temperature with Alexa Fluor-conjugated secondary antibodies. Hoechst 33342 (Sigma-Aldrich, B2261) was added at 1:1000 to counterstain nuclei. Slides were then washed and coverslipped. Antibody information is listed in Table S1. Images were captured with a Zeiss LSM700 confocal or a Zeiss Axio Imager.Z2 widefield microscope. Brightness and contrast were adjusted uniformly in Adobe Photoshop.

### Whole-mount immunostaining

Eyes were fixed in 4% PFA for 10 min at room temperature. Retinas were then isolated and post-fixed in 4% PFA for 1.5 h at 4°C. After two PBS washes, retinas were blocked in 3% donkey serum and 0.3% Triton X-100 in PBS for 2 h at room temperature before incubation with primary antibodies for 5 days at 4°C. Retinas were washed in PBS and incubated overnight at 4°C with secondary antibodies and Hoechst. Finally, retinas were washed, radially cut and mounted flat with antifade medium. Antibody information is listed in Table S1. Images were captured with a Zeiss LSM700 confocal, a Zeiss Axio Imager.Z2 widefield microscope or a Leica THUNDER Imager system. Confocal images were presented either as single optical sections or

as maximum-intensity projections of z-stacks. All images shown are representative of at least three biological replicates (n=3 mice per time point).

## Western blot analysis

Retinas were harvested from 2- to 3-month-old mice, snap-frozen in liquid nitrogen and minced with a needle. Tissues were then lysed in RIPA buffer (Sigma-Aldrich, R0278) supplemented with 1% phosphatase inhibitor cocktail (Wako, 167-24381) and 1% protease inhibitor cocktail (Sigma-Aldrich, P8340). The lysates were mixed with 3× SDS sample buffer containing 2-mercaptoethanol, boiled at 95°C for 5 min, and subjected to SDS-PAGE. Proteins were then transferred to a polyvinylidene difluoride (PVDF) membrane (Millipore, IPHV00010), blocked and immunoblotted with the following primary antibodies: NT5E/CD73 (D7F9A) rabbit mAb (1:1000, Cell Signaling, 13160; RRID: AB_2716625) and GAPDH (14C10) rabbit mAb (1:1000, Cell Signaling, 2118; RRID: AB_561053). Membranes were then incubated with anti-rabbit HRP-conjugated secondary antibodies (1:1000, Bio-Rad, 170-6515; RRID: AB_11125142) for 1 h at room temperature. For visualization, a chemiluminescence kit (Atto, 2332637) was used, and signals were detected using a WSE-6300 LuminoGraph III (Atto).

## Histological quantification

### Quantification of CD73CreER$^{T2}$-labeled cell density

For lineage-tracing experiments, mice (n≥3 per group) received tamoxifen at various time points ranging from E13.5 to 2 months of age (see Results section for details) and were all sacrificed at 3 months of age. Whole-mount retinas were imaged using a confocal microscope (10× objective, 0.5× digital zoom). To avoid overlap between central and peripheral regions, four central fields (near the optic nerve) and five peripheral fields (near the retinal edge) were captured per retina. To exclude the ONL, maximum-intensity projections of z-stacks were generated only from the INL to the GCL.

For quantification analysis, each of the four central fields was divided into two zones: 0-500 μm (C1) and 500-1000 μm (C2) from the optic nerve head. Likewise, each of the five peripheral fields was divided into two zones: 0-500 μm (P2) and 500-1000 μm (P1) from the retinal edge. In each zone of each field, tdTomato$^+$ astrocytes and tdTomato$^+$ cells in the INL were manually counted using the 'Cell Counter' tool in FIJI v2.16.0. The area of each zone was measured in the same software using the freehand selection tool. The density of tdTomato$^+$ cells was then calculated as the number of cells per 0.1 mm². Because multiple fields were obtained per mouse, the densities and area measurements from the same zone type (e.g. all C1 fields in one retina) were averaged to yield a single zone-based value per zone per mouse (i.e. one value for C1 per mouse, one for C2, etc.). For temporal analysis, these zone-averaged densities were averaged across mice at each tamoxifen administration time point, and, for spatial analysis, we plotted the zone-based densities (C1, C2, P1 and P2) for each tamoxifen injection time point.

All statistical analyses were performed in GraphPad Prism v10 (GraphPad Software). One-way ANOVA was used to compare tdTomato$^+$ cell densities among different time points (for temporal analysis) and among zones (C1, C2, P1 and P2; for spatial analysis), followed by Tukey's multiple comparisons test. Significance was defined as P<0.05. The exact sample size (n) is indicated in each figure legend, and results are expressed as mean±s.e.m., with each dot in the graphs representing data from an individual mouse.

## Vasculature radial length

We measured radial outgrowth as the distance from the optic nerve head to the vascular front. For each retina, we drew eight radial lines at approximately equal angles and measured the distance along each line in FIJI. Distances were recorded in μm. The eight values were averaged to yield one value per mouse.

## Filopodia counting

We adapted a published method for filopodia counting (Hellstrom et al., 2007). For each mouse, we acquired eight images of the vascular growth front (200 μm×200 μm) in the retina. In FIJI, we traced the continuous vascular front with a segmented line and measured its length. We then counted all filopodial protrusions that emerged from this leading edge within each field. Filopodia counts were normalized to the measured front length (number per 100 μm). Values from the eight fields were averaged to yield one value per mouse for statistics.

## Quantification of vascular morphology

We adapted a published Simple Neurite Tracer (SNT)-based method with minor modifications for quantification of vascular morphology (Shang and Schallek, 2024). We acquired 290.99 μm×290.99 μm fields from 500-1000 μm and 1000-1500 μm annuli centered on the optic nerve head. Z-stacks were restricted to the target plexus [deep plexus (DP) or intermediate plexus (IP)] and vessels were traced with the SNT plugin in FIJI. From SNT paths, we computed total cable length per field. We analyzed the following combinations: P10, DP at 500-1000 μm; P14, DP at 500-1000 μm and 1000-1500 μm; and 3 months of age, DP and IP at 500-1000 μm and 1000-1500 μm.

## Microglial quantification

We adapted a published method for microglial quantification with minor modifications (O'Koren et al., 2019). We acquired 290.99 μm×290.99 μm fields at P7-P14 and 640.17 μm×640.17 μm fields at 3 months of age at the outer plexiform layer (OPL) and counted Iba1$^+$ microglial cells. Acquisition areas were the 500-1000 μm and 1000-1500 μm annuli at P7, P10 and P14, or the 0-1250 μm and 1250-2500 μm annuli at 3 months of age, centered on the optic nerve head.

## Cell differentiation quantification

Retinal cell differentiation was examined at P14 using a modified protocol (Zhang et al., 2023). Briefly, 20 μm sections were prepared and immunolabeled with specific markers for retinal cell types (see Results section for details). Three sections per mouse were imaged (20× objective, 0.8× digital zoom; field size, 400.1 μm×400.1 μm) in both the superior and inferior retina at 500-1000 μm from the optic nerve. Cell counts were averaged for each mouse. All statistical analyses were performed using GraphPad Prism v10 (GraphPad Software) and exact n numbers are provided in the figure legends. Normality was assessed with the Shapiro–Wilk test, and groups were compared using an unpaired two-tailed Student's t-test (P<0.05 was considered statistically significant).

## Retinal morphometry

Retinal sections were obtained from wild-type and CD73 KO mice at P14, 1 month and 3 months of age (n≥7 per group). Sections were cut along the vertical meridian through the optic nerve and stained with H&E. ONL thickness was measured in 250 μm increments from the optic nerve head toward the retinal periphery (superior and inferior) using FIJI. Retinal morphometry graphs were generated using Origin 2024 (OriginLab). Statistical analyses were performed using R (version 4.4.2), and exact n numbers are provided in the figure legends. Normality was assessed using the Shapiro–Wilk test. At each age, ONL thickness at each eccentricity was compared between wild type and CD73 KO using an unpaired two-tailed Student's t-test, and P values across eccentricities were adjusted for multiple comparisons using the Benjamini–Hochberg false discovery rate (FDR) procedure (FDR-adjusted P-value <0.05 was considered statistically significant). Data are shown as mean±s.e.m.

## *In vivo* ERG

Mice at 2-3 months of age were dark-adapted overnight and the following procedures were performed under dim red light. The animals were anesthetized with midazolam (4 mg/kg), medetomidine (0.75 mg/kg) and butorphanol tartrate (5 mg/kg) before the recordings, as previously described (Miwa et al., 2019). The animal's pupils were dilated with eye drops of 0.5% tropicamide and 0.5% phenylephrine. The anesthetized animal was mounted on a stage with a heat pad inside a light-proof Faraday cage. A looped platinum recording electrode was placed on the cornea with a drop of 1.0% carboxymethyl cellulose and a reference electrode was positioned in the oral cavity. A ground electrode was clipped to the tail. Full-field light stimulus was delivered from a custom-made Ganzfeld dome. The stimulus, emitted from a green LED (M505L4, Thorlab), was attenuated using neutral density filters and transmitted to the Ganzfeld dome via a liquid light guide. For the photopic ERG, the mouse eye was light-adapted with background white light

(30 cd/cm$^2$). All mice were kept warm during the experiment using a heat pad. The photovoltage was amplified and filtered using a differential amplifier (bandpass: 0.5 to 300 Hz, Nihon Koden) and digitized at 10 kHz with Digidata1320A and AxoScope8. To analyze the amplitude of a- and b-wave, averaged responses were lowpass filtered below 70 Hz to eliminate oscillatory response beforehand. The implicit time (time to reach the peak amplitude) of the b-wave response evoked by dim light was estimated by fitting the response with the Gaussian function; $r(t) = R \cdot exp\left\{\frac{(t-\mu)^2}{2\sigma^2}\right\}$, where $R$ is the peak amplitude of the response, $\mu$ is the implicit time and $\sigma$ is the Gaussian width. The amplification constant of rod photoreceptors was estimated by fitting the initial phase of the a-wave with the equation proposed by Pugh and Lamb (1993).

### Rod single-cell recordings

For the single-cell recordings, the light-dependent current of rod photoreceptors was recorded by drowning the outer segments into a suction pipette as previously described (Sakurai et al., 2007). Briefly, a mouse aged 2-3 months was dark-adapted before the experiment. All the following procedures were performed under infrared light. The animals were euthanized by cervical dislocation and the dissociated retina was stored in equilibrated Locke's solution (112 mM NaCl, 3.6 mM KCl, 2.4 mM MgCl$_2$, 1.2 mM CaCl$_2$, 10 mM HEPES, 20 mM NaHCO$_3$, 3 mM Na$_2$-succinate, 0.5 mM Na-glutamate 10 mM glucose, 0.1% MEM vitamin and 0.1% MEM non-essential amino acids) before use at room temperature. The recording chamber was perfused with Locke's solution equilibrated with 95%O$_2$/5%CO$_2$ gas and preheated to 32-36°C. The recording electrode with an inter diameter adjusted to the width of the outer segment was prepared and filled with an electrode solution (140 mM NaCl, 3.6 mM KCl, 2.4 mM MgCl$_2$, 1.2 mM CaCl$_2$, 3 mM HEPES and 0.02 mM EDTA at pH 7.4). A piece of retina chopped with a razor blade in the electrode solution with 1.5 µg/ml DNaseI was transferred to the recording chamber mounted on the inverted microscope stage (IX73, Olympus). The photocurrent signal was amplified with an Axopatch 200B amplifier, low-pass filtered with a cutoff frequency of 30 Hz and digitized at 1 k Hz using Digidata1322A and pClamp8. The sensitivity ($I_{1/2}$) of photoreceptor was estimated by fitting the intensity-response data for each photoreceptor to the equation: $r = R_{max}\frac{I}{I+I_{1/2}}$, where $R_{max}$ is the maximal response amplitude (pA), $I$ is the flash intensity and $I_{1/2}$ is the sensitivity (photons/µm$^2$).

### scRNA-seq data analysis

Public single-cell RNA-seq datasets from GEO were re-analyzed (GSE118614, GSE175895 and GSE243413). No new sequencing was generated, and alignment/quantification pipelines were not re-run. We used the author-provided expression matrices and kept the original cell and cluster annotations. Cluster identities follow the source studies for GSE175895 (Zarkada et al., 2021) and GSE243413 (Li et al., 2024). For GSE118614, we adopted the cell-type labels of the author from the Goff lab GitHub repository (https://github.com/gofflab/developing_mouse_retina_scRNASeq). Trajectory analysis of the rod lineage was performed using Monocle3 (Cao et al., 2019), consistent with a previous study (Finkbeiner et al., 2022). Gene-level visualizations were performed using Seurat v5 (e.g. DimPlot, VlnPlot, DotPlot and FeaturePlot). Quantitative summaries and expression statistics were visualized with ggplot2 (tidyverse v2.0.0).

### Hypoxia/HIF signature scoring and correlation analysis

Similar to previous single-cell studies that applied module scores to gene signatures (Katoh et al., 2024; Liberzon et al., 2015), we calculated per-cell HIF-1 and HIF-2 signature scores from log-normalized Assay5 data. To obtain separate HIF-1 and HIF-2 scores, gene sets were defined from literature-curated HIF-1 (Benita et al., 2009) and HIF-2 (Kim et al., 2020) target lists. Scores were computed using Seurat v5 AddModuleScore with default binning and matched control features (Tirosh et al., 2016). In addition to individual HIF-1 and HIF-2 scores, we calculated a combined HIF signature (HIF-total) as the average of the HIF-1 and HIF-2 scores, and a HIF-bias score defined as (HIF1−HIF2)/(|HIF1|+|HIF2|+ε), where ε is a small constant to avoid division by zero. To assess the correlation between $Nt5e$ expression and HIF signatures along the rod pseudotime trajectory,

cells were ordered by Monocle3 pseudotime and grouped into 50 equidistant bins. For each bin, we calculated the average $Nt5e$ expression and the average HIF signature scores. Spearman's rank correlation coefficients were computed on these binned averages to mitigate single-cell noise. Resulting $P$ values were adjusted for multiple testing using the Benjamini–Hochberg false discovery rate (FDR) procedure, and the FDR-adjusted $P$ values are reported in Fig. S7.

### Statistical analysis

Statistical analyses were performed using GraphPad Prism v10 (GraphPad Software) or R (version 4.4.2). All data are reported as mean±s.e.m. Normality was assessed using the Shapiro–Wilk test. For comparisons between two groups, an unpaired two-tailed Student's $t$-test was used. For comparisons among three or more groups, one-way ANOVA was performed, followed by Tukey's multiple comparisons test. When appropriate, $P$ values were adjusted for multiple comparisons using the Benjamini–Hochberg false discovery rate (FDR) procedure, and significance was defined as $P<0.05$ (FDR-adjusted $P<0.05$ for analyses with multiple comparisons).

### Acknowledgements

We thank Dr Satoru Takahashi, Dr Erna Raja and Dr Marina Sanaki-Matsumiya for their experimental support, and Dr Keiichi Asano for his advice on scRNA-seq re-analysis. We also thank that Dr Fuminori Tsuruta, Dr Yuya Sanaki, and Dr Masafumi Muratani for critically reading the manuscript. We are grateful to the Laboratory Animal Resource Center at the University of Tsukuba for their professional care of the animals and support with mouse husbandry and colony maintenance, as well as to Ms Mariko Higashi, Ms Mami Ho, Ms Tomomi Zama and Ms Eri Motoyama for their technical assistance.

### Competing interests

The authors declare no competing or financial interests.

### Author contributions

Conceptualization: R.I.; Formal analysis: R.I., K.S.; Funding acquisition: R.I., K.K.; Investigation: R.I., K.S., N.H., S.M., K.K.; Project administration: R.I.; Supervision: R.I., H.Y.; Visualization: R.I.; Writing – original draft: R.I., K.S.; Writing – review & editing: R.I., K.S., B.K.F., S.M., K.K., H.Y.

### Funding

This study was supported in part by the Japan Society for the Promotion of Science [KAKENHI grant numbers JP22K16940 (to R.I.) and JP22K09346], the Japan Agency for Medical Research and Development (grant number JP23bm1123032), by the Terumo Foundation for Life Sciences and Arts, by the Uehara Memorial Foundation, by the NOVARTIS Foundation, by the Inamori Foundation, by the SENSHIN Medical Research Foundation and by the Takeda Science Foundation (to K.K.). Open Access funding provided by the Takeda Science Foundation. Deposited in PMC for immediate release.

### Data and resource availability

All relevant data and details of resources can be found within the article and its supplementary information.

### Peer review history

The peer review history is available online at https://journals.biologists.com/dev/lookup/doi/10.1242/dev.205013.reviewer-comments.pdf.

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
