## [Peer Review File · Development (Cambridge, England)]

Spatiotemporal dynamics of ecto-5'-nucleotidase (CD73) in mouse retina under physiological conditions

Ryutaro Ishii, Keisuke Sakurai, Nao Hosomi, Bernd K. Fleischmann, Seiya Mizuno, Kenichi Kimura and Hiromi Yanagisawa
DOI: 10.1242/dev.205013

Editor: François Guillemot

Review timeline

Original submission:	9 June 2025
Editorial decision:	16 August 2025
First revision received:	12 December 2025
Accepted:	29 December 2025

Original submission

First decision letter

MS ID#: dev.205013

MS Title: Spatiotemporal dynamics of CD73 in mouse retina under physiological conditions

Authors: Keisuke Sakurai; Nao Hosomi; Bernd K. Fleischmann; Seiya Mizuno; Kenichi Kimura; Hiromi Yanagisawa; Ryutaro Ishii
Article Type: Research Article

Dear Dr Ishii,

I have now received the reports of three referees on your manuscript and I have reached a decision. The referees' comments are appended below, or you can access them online: please go to: *****

As you will see, the referees express great interest in your work, but they also have significant criticisms and recommend a substantial revision of your manuscript before we can consider publication. If you are able to revise the manuscript along the lines suggested, which may involve further experiments, I will be happy to receive a revised version of the manuscript. Please note that some of the experiments requested by reviewer 3 seem outside the scope of the manuscript and are not necessary for the revision of the paper. Your revised paper will be re-reviewed by one or more of the original referees, and acceptance of your manuscript will depend on your addressing satisfactorily the reviewers' major concerns. Please also note that Development will normally permit only one round of major revision.

If it would be helpful, you are welcome to contact us to discuss your revision in greater detail. Please send us a point-by-point response indicating your plans for addressing the referees' comments, and we will look over this and provide further guidance.

Please attend to all of the reviewers' comments and ensure that you clearly highlight all changes made in the revised manuscript. Please avoid using 'Tracked changes' in Word files as these are lost in PDF conversion. I should be grateful if you would also provide a point-by-point response detailing how you have dealt with the points raised by the reviewers in the 'Response to Reviewers' box. If you do not agree with any of their criticisms or suggestions please explain clearly why this is so.

Reviewer 1

SUMMARY OF THE ADVANCE MADE IN THIS PAPER AND ITS POTENTIAL SIGNIFICANCE TO THE FIELD

The gene CD73 has previously been described to be a marker of rod photoreceptors. As a gene that can regulate adenosine secretion, CD73 can also regulate cell signaling, and prior works have demonstrated that CD73 is required for developmental processes.

Here, Ishii et al. deploy two transgenic mouse lines in order to characterize CD73 expression during retinal development. They use a GFP BAC reporter allele to examine CD73 expression, and then deploy a CD73-CreER line to fate map CD73-expressing cells. Taking advantage of the fact that the latter allele is a knock-in, they also generate homozygotes in order to determine the genetic requirement for CD73 in retinal development. Using electroretinography, they describe a requirement for CD73 (and therefore adenosine secretion) in retinal electrophysiology. The quality of the data are very high, and I have only a few minor criticisms about the study. Moreover, the paper is very well written, and well researched. The text is thought provoking, and the portrayal is well balanced.

The main issue for me is that the novelty and impact of the work is of limited scope. I also think that the phenotypic data would be of greater interest to a different audience. Understanding the ontogeny of CD73 is of importance to the field, and while the current manuscript would constitute the definitive study of retinal CD73 expression, it is not the first to report such profiling. The work tracks CD73 very well, but does not really provide much insight into questions of retinal development, retinal cell lineage, migration, or other developmental dynamics. In my opinion, the phenotypic data will be of more interest to an audience interested in physiology than Development.

SUGGESTIONS TO AUTHORS

The cell count data suggest that CD73 is very well expressed in rod photoreceptors as one might have expected. Data on other cell types, including Muller glia, astrocyte precursors, and amacrine, suggest that a subset of cells are labelled. Why would this be the case? Are there particular subtypes of these cells that express CD73, or are CD73 levels very low in these cells? Further investigation (e.g. antibody staining, scRNA-seq analyses) could be used to determine whether CD73 expression is variable in these cell types, or whether the results reflect low-level expression by all cells of a given class, and then sporadic marking via the Cre strategy. Can BAC-GFP expression be seen in these cells too? If not, does that suggest that the expression of CD73 is very transient and/or low-level?

I think another detail that might help improve the impact of the study, might be to describe the expression dynamics of adenosine receptors during retinal development. It would be of interest to understand how signalling initiated by CD73 would be received. Are there developmental phenotypes in receptor knockouts that might be relevant to discuss or address experimentally?

Minor comments:

- 1) Please note that Vsx2 also marks retinal progenitor cells (in addition to bipolars).
- 2) The retinal thickness phenotype was not convincing to me. If I am interpreting the data correctly, there are 32 t tests being performed in Fig. S10. By chance, at least one such test will hit significance. It would be more convincing to see the data analyzed using a statistical method that controls for false-discovery. Retinal thickness in adult stages varies via a process called emmetropization, which has mainly been linked to Muller glia so far. This is a minor point, but more mechanistic insight into this potential phenotype would be of interest.

Reviewer 2

SUMMARY OF THE ADVANCE MADE IN THIS PAPER AND ITS POTENTIAL SIGNIFICANCE TO THE FIELD

Excellent advance in understanding CD73 function in the retina.

This manuscript describes spatiotemporal dynamics of CD73 using transgenic mouse lines (CD73-BAC-EGFP and CD73-CreERT2) and examines CD73 expression in the rod lineage and in the inner nuclear layer. The authors demonstrate that CD73 is important for rod photoresponse using CD73 deletion mice and suggest its potential role in light/dark adaptation. Overall, this is a clean and straightforward manuscript. The data are excellent, and conclusions are justified. Immunostainings and quantifications are performed well. The following suggestions will further improve this excellent paper.

1. The authors begin the Abstract and the Intro with Adenosine signaling in the retina and try to establish why they are focusing on the dynamics of ecto-5'-nucleotidase (i.e., CD73), which is the enzyme involved in extracellular adenosine generation. However, the title is somewhat low key and doesn't allow one to appreciate the significance of the study. The authors should include what CD73 is in the title.
2. The introduction can be improved for clarity and coherence. It does not fully establish the significance or novelty of the study and is somewhat dense covering too many concepts.
3. It would be interesting if the authors can check published single cell or high-resolution spatial transcriptomic data from developing mouse and/or human retina for expression of CD73. They may also want to check scRNA-seq data in Campello et al (PMID: 39954235) for CD73 expression in rod subtypes and during aging. It was unclear how many rods were used for single cell recordings and whether there was some heterogeneity, which can be explained by two rod subtypes.
4. Age of the mice used for IHC or ERG should be mentioned in the Figure Legends. This reviewer had to look at the methods to find out what was being done when.
5. The authors might consider discussing how temporal expression changes align with developmental milestones.
6. The authors might consider temporal bulk and scRNA-seq profiling of Cd73-ko retina. This may provide mechanistic insights on the role of CD73.
7. Figure 8 is mentioned in the Discussion. The figure provides a nice summary of the proposed model but needs better description. Can also be used as a graphical abstract.

Reviewer 3

Ryutaro Ishii and his colleagues investigated expression pattern of CD73 in the retina utilizing two types of transgenic mice. While some of their findings may be novel and interesting, they are far from supporting the conclusions drawn in the current approach. Additional critical experiments are needed to achieve their goal.

Major points

1. Basically, the current experiments have only demonstrated the expression pattern of CD73 at certain stages of retinal development. However, in the abstract and discussion, the authors state that physiological cues such as hypoxia regulate CD73 expression as an upstream factor. It is not possible to conclude a causal relationship without appropriate interventions. To investigate the roles of oxygen supplementation in the retina, oxygen-induced retinopathy (OIR) model is often used. In fact, the authors referred to this model by citing relevant literature. At this stage, performing the OIR challenge is essential to support their conclusion. Co-staining for HIF-1 α or HIF-2 α in the transgenic mice is also necessary, as they discussed.
2. They also mentioned light as a regulator for CD73 expression and adenosine metabolism. To confirm the role of the light, interventional experiments such as constant darkness or light modulation/obstruction would be necessary.
3. Vascular development is crucial in the retina. This is particularly important given that CD73 is thought to be involved in oxygen supply and metabolism. In CD73 knockout mice, not only ERG but also the vascular pattern should be investigated in examined.
4. They mention inflammation at some point in the manuscript. The distribution of inflammatory cells, such as microglia and macrophages, should be examined in the knockout mice.

Minor points

1. In the abstract, they state "dysregulation of adenosine signalling contributes to retinal diseases (line 35)" and "disease-linked alterations in CD73 (line 46)." It would be helpful to list, in the introduction, the retinal diseases known to be associated with adenosine signaling and CD73.
2. They state "Therefore, in the adult retina, adenosine levels fluctuate in response to daily physiological cues (line 72)". Are adenosine levels measurable in this study? If not, please speculate on and discuss the potential changes of adenosine levels.
3. Please provide a more detailed explanation of "retinal waves (line 81)".
4. impact on adenosine signalling (line 97)
5. The term "retinal cell differentiation" would be more appropriate than "retinal cell genesis" (Line 237).
6. ONL was significantly thicker in CD73 KO mice (line 307). How could this occur? Please discuss the possible explanations in more detail.
7. They conclude, "uncover new therapeutic targets for conditions in which lifestyle factors drive inflammation and compromise vascular integrity, such as diabetic retinopathy and age-related macular degeneration" (Line 474). How is the the current study relevant to diabetic retinopathy or age-related macular degeneration? This point could probably be discussed further in a dedicated paragraph.

First revision

Author response to reviewers' comments

Spatiotemporal dynamics of CD73 in mouse retina under physiological conditions
Response to the editor and reviewers

We would like to thank the editors and reviewers for their constructive criticism and insightful suggestions. Below, please find point-by-point responses to each comment.

Editor

As you will see, the referees express great interest in your work, but they also have significant criticisms and recommend a substantial revision of your manuscript before we can consider publication. If you are able to revise the manuscript along the lines suggested, which may involve further experiments, I will be happy to receive a revised version of the manuscript. Please note that some of the experiments requested by reviewer 3 seem outside the scope of the manuscript and are not necessary for the revision of the paper. Your revised paper will be re-reviewed by one or more of the original referees, and acceptance of your manuscript will depend on your addressing satisfactorily the reviewers' major concerns. Please also note that Development will normally permit only one round of major revision.

Thank you for handling our manuscript and for the constructive guidance in your decision letter. In accordance with your recommendation, we have prepared a substantially revised version that addresses the referees' major concerns within this single round of major revision. In this revision, we added analyses of retinal vasculature and microglial phenotypes in CD73 knock-in/knockout (CD73 KO) mice (see Fig. S16, and S17, Page 12, Lines, 250-259; Fig S19, Pages 13, Lines 266-270) and re-analyzed publicly available single-cell RNA-seq (scRNA-seq) datasets to support our in vivo CD73-BAC-EGFP (CD73-EGFP) and *CD73-CreER^{T2}* lineage-tracing results (see Fig. S6B, Page 9, Lines 162-164; Fig S7, Page 9, Lines 172-176; Figs. S6B, S9, and S10, Page 10, Lines 198-205; Fig. S14,

Page 12, Lines 241-246; Fig. S18A, Page 12-13, Lines 262-264). Below, we provide a detailed point-by-point response to each reviewer.

Reviewer 1

The main issue for me is that the novelty and impact of the work is of limited scope. I also think that the phenotypic data would be of greater interest to a different audience. Understanding the ontogeny of CD73 is of importance to the field, and while the current manuscript would constitute the definitive study of retinal CD73 expression, it is not the first to report such profiling. The work tracks CD73 very well, but does not really provide much insight into questions of retinal development, retinal cell lineage, migration, or other developmental dynamics. In my opinion, the phenotypic data will be of more interest to an audience interested in physiology than Development.

We thank the reviewer for the thoughtful comments. To better situate CD73 within retinal development, in this revision we surveyed the expression of *Nt5e* (CD73), adenosine receptor genes, and *P2ry12* across developmental and adult scRNA-seq datasets (see Fig. S7, Page 9, Lines 172-176; Figs. S6B, S9, and S10, Page 10, Lines 198-205; Fig. S14, Page 12, Lines 238-246; Fig. S18A, Page 12-13, Lines 262-264). In addition, we added quantitative analyses of the retinal vasculature and microglia in CD73 KO mice (see Fig. S16, and S17, Page 12, Lines, 250-259; Fig S19, Pages 13, Lines 266-269). These additions strengthen our spatiotemporal mapping and better align the manuscript with the scope of Development.

The cell count data suggest that CD73 is very well expressed in rod photoreceptors as one might have expected. Data on other cell types, including Muller glia, astrocyte precursors, and amacrine, suggest that a subset of cells are labelled. Why would this be the case? Are there particular subtypes of these cells that express CD73, or are CD73 levels very low in these cells? Further investigation (e.g. antibody staining, scRNA-seq analyses) could be used to determine whether CD73 expression is variable in these cell types, or whether the results reflect low-level expression by all cells of a given class, and then sporadic marking via the Cre strategy. Can BAC-GFP expression be seen in these cells too? If not, does that suggest that the expression of CD73 is very transient and/or low-level?

We agree with the reviewer's interpretation. CD73-EGFP signals in the INL and in astrocyte precursors are faint but present, and EGFP⁺ cells are rare, consistent with *CD73-CreER^{T2}* lineage tracing (see Fig. S4, Page 8, Lines 143-144). Prior scRNA-seq work has also mentioned the possibility of *Nt5e* transcripts in the INL (Li et al.; PMID: 38812536). To examine this in more detail, we re-analyzed *Nt5e* expression in the INL using that public scRNA-seq datasets and detected *Nt5e* across all major INL classes (Müller glia, amacrine, and horizontals) (see Fig. S6B, Page 10, Lines 198-203). Taken together, these data indicate that the sparse labeling is at least partly biological rather than a reporter artifact. Moreover, by examining the *Nt5e* expression in the cone bipolar cell subtypes and amacrine cell subtypes (see Fig. S9B-D, S10; Pages 10, Lines 201-203), we found that *Nt5e* expression was not obviously enriched in any subtype. These results support the idea that *Nt5e* expression in the INL is widespread rather than restricted to specific subtypes. At the same time, because *Nt5e* transcript counts are low, we still cannot distinguish whether this pattern reflects transient expression or stable but low-level expression. We have added a description of this limitation to the revised manuscript in the Discussion (see Page 19, Lines 406-408).

I think another detail that might help improve the impact of the study, might be to describe the expression dynamics of adenosine receptors during retinal development. It would be of interest to understand how signalling initiated by CD73 would be received. Are there developmental phenotypes in receptor knockouts that might be relevant to discuss or address experimentally?

We thank the reviewer for the helpful suggestion. We stated more explicitly that previous studies suggested A1R and A2AR knockouts show no abnormalities in normal retinal vascular development under physiological conditions (see Page 12, Lines 247-248; Page 21, Line 457-459). In addition, we now briefly describe developmental adenosine receptor expression based on public scRNA-seq datasets in the Results (see Fig. S14, Page 12, Line 241-245).

Minor Comments

Please note that *Vsx2* also marks retinal progenitor cells (in addition to bipolars).

We thank the reviewer for the clarification. We have revised the text to state that *Vsx2* marks retinal progenitor cells as well as bipolar cells (see Fig. 1E, Page 8, Line 133).

The retinal thickness phenotype was not convincing to me. If I am interpreting the data correctly, there are 32 t tests being performed in Fig. S10. By chance, at least one such test will hit significance. It would be more convincing to see the data analyzed using a statistical method that controls for false-discovery. Retinal thickness in adult stages varies via a process called emmetropization, which has mainly been linked to Muller glia so far. This is a minor point, but more mechanistic insight into this potential phenotype would be of interest.

We appreciate this important comment. As you mentioned, emmetropization plays a key role in retinal growth. Therefore, to reduce the risk of false discovery, we reanalyzed ONL thickness at 1 and 3 months of age using FDR correction across eccentricities and found no significant genotype differences at 3 months (see Fig. S22A, Page 15, Lines 338-342). In line with your helpful suggestion, we also added a note in the Discussion that adult retinal thickness can vary as part of the emmetropization process and we now state that our data do not support a robust phenotype in adult CD73 KO mice (see Page 21, Lines 466-469).

Reviewer 2

Overall, this is a clean and straightforward manuscript. The data are excellent, and conclusions are justified. Immunostainings and quantifications are performed well. The following suggestions will further improve this excellent paper.

We would like to thank the reviewer for their positive comments. We address each point below.

1. The authors begin the Abstract and the Intro with Adenosine signaling in the retina and try to establish why they are focusing on the dynamics of ecto-5'-nucleotidase (i.e., CD73), which is the enzyme involved in extracellular adenosine generation. However, the title is somewhat low key and doesn't allow one to appreciate the significance of the study. The authors should include what CD73 is in the title.

Thank you for the critical comments on this article. We modified the title; Spatiotemporal dynamics of ecto-5'-nucleotidase (CD73) in mouse retina under physiological conditions (see Page 1, Line 1).

2. The introduction can be improved for clarity and coherence. It does not fully establish the significance or novelty of the study and is somewhat dense covering too many concepts.

We appreciate this critical suggestion. In the revised Introduction, we reduced unnecessary detail and reorganized the text as follows: (1) a brief overview of adenosine in the central nervous system; (2-3) key features of adenosine metabolism and signaling in the retina; and (4) an outline of the current gap between what is known about CD73 and retinal diseases.

3. It would be interesting if the authors can check published single cell or high-resolution spatial transcriptomic data from developing mouse and/or human retina for expression of CD73. They may also want to check scRNA-seq data in Campello et al (PMID: 39954235) for CD73 expression in rod subtypes and during aging. It was unclear how many rods were used for single cell recordings and whether there was some heterogeneity, which can be explained by two rod subtypes.

We thank the reviewer for the valuable suggestion. Our study focuses on developmental and young adult stages up to 3 months rather than aging, and adding further investigation of aging or additional electrophysiological experiments may fall outside the main scope of this journal, in line with reviewer 1's suggestion. Although rods are a major site of CD73 expression, the main novelty

of this study lies in the finding that CD73 is expressed in the retinal cell types other than rods, as revealed by two new transgenic mouse lines. Therefore, in this manuscript we have only added that the sample sizes (n) for single-cell recordings indicate the number of rods in the figure legend (see Page 49, Lines 1136-1151). Instead, to support our findings on CD73 expression, we examined published scRNA-seq datasets. This analysis revealed *Nt5e* expression in the RPE, as well as in INL cell types, which supports the presence of tdTomato⁺ RPE cells in our lineage-tracing experiments. We have added this result to the Result section (see Fig. S6B, Page 9, Lines 162-164).

4. Age of the mice used for IHC or ERG should be mentioned in the Figure Legends. This reviewer had to look at the methods to find out what was being done when.

Thank you for pointing this out. We have added the ages of the mice for all IHC and ERG experiments to the figure legends (see Page 46, Line 1071; Page 47, Line 1093; Page 48, Line 1111; Page 49, Line 1131, and Line 1151).

5. The authors might consider discussing how temporal expression changes align with developmental milestones.

We thank the reviewer for this suggestion. We have restructured the Discussion into specific subsections to clarify the alignment with developmental milestones: Hypoxia (Pages 17, Lines 360-385), and retinal waves/light (Pages 18-19, Line 387-424). Additionally, we have added a new schematic figure illustrating developmental milestones, including retinal wave, light, and hypoxia (see Fig. S23, Page 17, Lines 355-358).

6. The authors might consider temporal bulk and scRNA-seq profiling of Cd73-ko retina. This may provide mechanistic insights on the role of CD73.

We appreciate this valuable suggestion and agree that bulk and scRNA-seq of CD73 KO retinas would provide mechanistic insight. Because of time and scope limitations for the present revision, we instead reanalyzed public scRNA-seq datasets. In line with Reviewer 3's suggestion to investigate the vascular phenotype of CD73 KO mice, we surveyed adenosine receptor expression in vascular compartments using a public scRNA-seq dataset (PMID: 34273276). This analysis shows that, among adenosine receptor genes, *Adora2a* (A2AR) is enriched in endothelial cells, especially tip cells, at P6 and P10. In the adult retina, both endothelial cells and pericytes express *Adora2a* (see Fig. S14, Page 12, Lines 242-246). Consistent with these expression patterns, the total cable length of the deep plexus at P10 was significantly higher in CD73 KO than in WT mice, and CD73 KO mice showed a reduction in deep and intermediate plexus cable length at 3 months of age (see Fig. S17, Page 12, Lines 235-257). Taken together, these observations support a working model in which CD73 helps maintain adenosine tone that modulates the balance of endothelial dynamics under physiological conditions.

7. Figure 8 is mentioned in the Discussion. The figure provides a nice summary of the proposed model but needs better description. Can also be used as a graphical abstract.

We thank the reviewer for this helpful comment. We have added labels to Figure 8 (see Fig. 8). In addition, we added a milestones figure summarizing important events during development to clearly link CD73 expression and signaling to each stage, facilitating the understanding of our hypothesis (see Fig. S23, Page 17, Lines 355-358).

Reviewer 3

1. Basically, the current experiments have only demonstrated the expression pattern of CD73 at certain stages of retinal development. However, in the abstract and discussion, the authors state that physiological cues such as hypoxia regulate CD73 expression as an upstream factor. It is not possible to conclude a causal relationship without appropriate interventions. To investigate the roles of oxygen supplementation in the retina, oxygen-induced retinopathy (OIR) model is often used. In fact, the authors referred to this model by citing relevant literature. At this stage, performing the OIR challenge is essential to support their conclusion. Co-staining for HIF-1 α or HIF-2 α in the transgenic mice is also necessary, as they discussed.

We appreciate this valuable suggestion and agree that mechanistic interventions would provide definitive causal tests. We actually attempted immunostaining for HIF-1 α and HIF-2 α ; however, under our conditions, the signals were not robust enough for reliable interpretation. Since the present study focuses on physiological conditions, we prioritized examining the relationship between HIF activity and *Nt5e* by analyzing HIF-related gene signatures in public scRNA-seq datasets, rather than performing OIR interventions. Along the rod lineage pseudotime, the HIF-1-biased signature was positively associated with *Nt5e* expression. We present these observations as hypothesis-generating, soften the sentence in Abstract (see Page 3, Line 45-47) and Discussion (see Page 21, Line 476-478), and note that an OIR challenge would be an appropriate future test to establish causality. We have replaced the previous discussion based on literature with a description of these new results (see Fig. S7, Page 9, Lines 172-176; Page 17, Lines 363-367).

2. They also mentioned light as a regulator for CD73 expression and adenosine metabolism. To confirm the role of the light, interventional experiments such as constant darkness or light modulation/obstruction would be necessary.

We thank the reviewer for this important comment. Because our present study focuses on characterizing CD73 under physiological light/dark conditions, we did not add interventional experiments such as constant darkness or light obstruction, in line with the editor's suggestion. Instead, we added a developmental milestones figure to state explicitly that our ideas about light regulation are presented as hypotheses based on observations under normal light/dark conditions (see Fig. S23, Page 17, Lines 355-358).

3. Vascular development is crucial in the retina. This is particularly important given that CD73 is thought to be involved in oxygen supply and metabolism. In CD73 knockout mice, not only ERG but also the vascular pattern should be investigated in examined.

We thank the reviewer for this valuable comment. In response, we assessed retinal vascular development under physiological conditions in CD73 KO and WT mice. Consistent with prior reports under physiological conditions (PMID: 36029802), we observed no overt abnormalities in superficial vascularization (see Figs. S16, Page 12, Lines 252-253). However, quantitative analysis of vascular morphology in the 500-1000 μ m and 1000-1500 μ m annuli from the optic nerve head revealed increased total cable length in the deep plexus at P10 in the 500-1000 μ m regions of CD73 KO retinas, and reduced total cable length in the deep and intermediate plexuses at 3 months of age in the same region (see Fig. S17, Page 12, Lines 253-257). As shown by our scRNA-seq analysis, endothelial cells and pericytes express adenosine receptors (see Fig. S14, Page 12, Lines 241-246).

4. They mention inflammation at some point in the manuscript. The distribution of inflammatory cells, such as microglia and macrophages, should be examined in the knockout mice.

We thank the reviewer for this helpful suggestion. To assess the inflammatory status under physiological conditions, we first re-analyzed public scRNA-seq data (PMID: 38812536). In this dataset, high levels of *P2ry12*, and among adenosine receptors, *Adora3* (see Fig. S18A, Page 12-13, Lines 262-264) were higher than those of any adenosine receptor, and among adenosine receptors *Adora3* showed the highest levels. Immunostaining confirmed that P2RY12 expression in Iba1⁺ (see Fig. S18B, Page 13, Line 265), and quantification from development through 3 months of age showed only a weak, non-significant trend toward higher Iba1⁺ cell density in CD73 KO mice (see Fig. S19, Page 13, Lines 266-269).

Minor Comments

1. In the abstract, they state "dysregulation of adenosine signalling contributes to retinal diseases (line 35)" and "disease-linked alterations in CD73 (line 46)." It would be helpful to list, in the introduction, the retinal diseases known to be associated with adenosine signaling and CD73.

We thank the reviewer for this helpful suggestion. Considering a concurrent request to simplify the Introduction, we added a sentence succinctly referencing retinal diseases associated with adenosine metabolism (PMID: 32109489, and 37696869) (see Page 5, Lines 89-91).

2. They state "Therefore, in the adult retina, adenosine levels fluctuate in response to daily physiological cues (line 72)". Are adenosine levels measurable in this study? If not, please speculate on and discuss the potential changes of adenosine levels.

We thank the reviewer for this helpful suggestion. We clarified that we did not directly measure retinal adenosine levels in this study (see Pages 20, Line 444). Accordingly, we revised the paragraph to discuss hypothesized adenosine changes based on prior literature (PMID: 15634784, and 33510619).

3. Please provide a more detailed explanation of "retinal waves (line 81)".

We thank the reviewer for this helpful suggestion. Guided by the review cited in our manuscript, we added a brief sentence explaining retinal waves in the Introduction (see Page 4, Lines 77-80).

4. impact on adenosine signalling (line 97)

We thank the reviewer for this helpful suggestion. Based on a brief overview of adenosine metabolism, we clarified that CD73 acts as a pivotal trigger for adenosine signaling by regulating extracellular adenosine levels (see Page 5, Lines 91-94).

5. The term "retinal cell differentiation" would be more appropriate than "retinal cell genesis" (Line 237).

We thank the reviewer for this helpful suggestion. We have replaced "retinal cell genesis" with "retinal cell differentiation." (see Page 13, Line 274, and 277; Page 14, Line 289; Page 28, Lines 636-637).

6. ONL was significantly thicker in CD73 KO mice (line 307). How could this occur? Please discuss the possible explanations in more detail.

We thank the reviewer for this helpful suggestion. As described in our response to Reviewer 1, after applying FDR correction we found no significant differences in ONL thickness between WT and CD73 KO mice. We now describe this only as a mild, non-significant trend and note that adult retinal thickness can vary during emmetropization (see Fig. S22, Page 15-16, Lines 338-342; Page 21, Lines 466-469).

7. They conclude, "uncover new therapeutic targets for conditions in which lifestyle factors drive inflammation and compromise vascular integrity, such as diabetic retinopathy and age-related macular degeneration" (Line 474). How is the the current study relevant to diabetic retinopathy or age-related macular degeneration? This point could probably be discussed further in a dedicated paragraph.

We thank the reviewer for this helpful suggestion. We have added a paragraph to outline how altered CD73 dependent adenosine metabolism and vascular homeostasis may be relevant to diabetic retinopathy and age-related macular degeneration, with supporting citations including human genetics studies (see Page 20-21, Lines 457-478). We also revised the Abstract and the concluding sentence of the Discussion to avoid overstatement by referring to genetic and environmental factors (see Page 3, Line 47; Page 6, Line 104).

Second decision letter

MS ID#: dev.205013R1

MS Title: Spatiotemporal dynamics of ecto-5'-nucleotidase (CD73) in mouse retina under physiological conditions

Authors: Keisuke Sakurai; Nao Hosomi; Bernd K. Fleischmann; Seiya Mizuno; Kenichi Kimura; Hiromi Yanagisawa; Ryutaro Ishii
Article Type: Research Article

Dear Dr Ishii,

I am happy to tell you that your manuscript has been accepted for publication in Development, pending our standard publication integrity checks.

It was accepted on 29 Dec 2025. As you will see from the referees' reports appended below, two of the three referees are against publication of the revised manuscript. However, after reading their reports and re-examining the manuscript, I found that your work has sufficient merit to be published in Development and I have therefore accepted the manuscript.

Reviewer 1

SUMMARY OF THE ADVANCE MADE IN THIS PAPER AND ITS POTENTIAL SIGNIFICANCE TO THE FIELD

The gene CD73 has previously been described to be a marker of rod photoreceptors. As a gene that can regulate adenosine secretion, CD73 can also regulate cell signaling, and prior works have demonstrated that CD73 is required for developmental processes.

Here, Ishii et al. deploy two transgenic mouse lines in order to characterize CD73 expression during retinal development. They use a GFP BAC reporter allele to examine CD73 expression, and then deploy a CD73-CreER line to fate map CD73-expressing cells. Taking advantage of the fact that the latter allele is a knock-in, they also generate homozygotes in order to determine the genetic requirement for CD73 in retinal development. Using electroretinography, they describe a requirement for CD73 (and therefore adenosine secretion) in retinal electrophysiology. In the revision, the authors also examine the development of the retinal vasculature. The quality of the data are very high, and I have only a few minor criticisms about the study. Moreover, the paper is very well written, and well researched. The text is thought provoking, and the portrayal is well balanced.

Despite the high quality of the work, I did not feel that the major concerns raised by myself and Reviewer 3 were addressed in a meaningful way. I appreciated the efforts to address the reviewer comments, but I still find that the paper does not meaningfully advance our understanding of eye development very far beyond existing works. For me, the concepts advanced by the paper are limited in scope.

The following quote is taken verbatim from the 'Instructions for Reviewers' for Development. "The main criterion for publication in Development is that a Research Article or Report should make a significant and novel contribution to our understanding of developmental mechanisms, and should be of broad interest to the developmental biology and/or stem cell community. Studies lacking such a contribution, no matter how meticulous, are not acceptable for publication..." I find that the current manuscript is a meticulous and very well performed study that does not have a novel conceptual advance related to the development of the eye. As such, I find that it does not meet the main stated criterion necessary for publication at Development. I think that the paper is superbly well-performed, but that its main conceptual advances are related to retinal physiology rather than developmental mechanisms. The reported relationships between CD73 expression and various cell types already has precedence in the literature, so I find that the main findings of the paper are very well done, but incremental in this sense.

I interpreted the comments of Reviewer 3 as pointing out the same shortcoming in a different way. Reviewer 3 suggested several experiments that would advance the scope of the paper towards a conceptual advance that would be required for publication at Development. I think that the Authors have made attempts to address the concerns pointed out by myself and Reviewer 3, but

the provided revision circles back over the same models rather than extending the scope of the paper towards novelty. They have added a lot of scRNA-seq data from wild-type mice. These new data improve the paper (which was already excellent so far as it goes), but do little to advance its overall scope unfortunately. Ultimately, I still feel that the paper should be published, and will make an excellent contribution to the field, but is simply not a good fit for Development, where a clear conceptual advance related to developmental biology is required. I appreciate the attempt to address the minor concerns that were raised, but the major concerns remain in my opinion. I am sorry to be negative about this, but I think that the criteria for publication at Development are very clear.

Reviewer 2

SUMMARY OF THE ADVANCE MADE IN THIS PAPER AND ITS POTENTIAL SIGNIFICANCE TO THE FIELD

SUGGESTIONS TO AUTHORS

The authors have appropriately responded to the reviewers' comments.

Reviewer 3

Unfortunately, the authors did not adequately address my previous suggestions, particularly those regarding interventional approaches to establish a causal relationship. Therefore, I am unable to provide further substantive comments on the revised manuscript.